# Learning Informative Latent Representation for Quantum State Tomography

## Abstract

Quantum state tomography (QST) is the process of reconstructing the complete state of a quantum system (mathematically described as a density matrix) through a series of different measurements. These measurements are performed on a number of identical copies of the quantum system, with outcomes gathered as probabilities/frequencies. QST aims to recover the density matrix and the corresponding properties of the quantum state from the measured frequencies. Although an informationally complete set of measurements can specify the quantum state accurately in an ideal scenario with a large number of identical copies, both the measurements and identical copies are restricted and imperfect in practical scenarios, making QST highly ill-posed. The conventional QST methods usually assume adequate or accurate measured frequencies or rely on manually designed regularizers to handle the ill-posed reconstruction problem, suffering from limited applications in realistic scenarios. Recent advances in deep neural networks (DNNs) led to the emergence of deep learning (DL) in QST. However, existing DL-based QST approaches often employ generic DNN models that are not optimized for imperfect conditions of QST. In this paper, we propose a transformer-based autoencoder architecture tailored for QST with imperfect measurement data. Our method leverages a transformer-based encoder to extract *an informative latent representation* (ILR) from imperfect measurement data and employs a decoder to predict the quantum states based on the ILR. We anticipate that the high-dimensional ILR will capture more comprehensive information about the quantum states. To achieve this, we conduct pre-training of the encoder using a pretext task that involves reconstructing high-quality frequencies from measured frequencies. Extensive simulations and experiments demonstrate the remarkable ability of the informative latent representation to deal with imperfect measurement data in QST.

## 1 Introduction

The determination of the quantum state of a system, known as quantum state tomography (QST), is the gold standard for verification and benchmarking of quantum devices and is also essential for engineering systems in developing quantum technology (Gebhart et al., 2023; Rambach et al., 2021). In the physical world, the full description of a quantum state can be represented as a density matrix $\rho$, i.e., a positive-definite Hermitian matrix with unit trace $\rho = \rho^{\dagger}, \mathrm{Tr}(\rho) = 1$. To determine $\rho$, one may first perform measurements on a collection of identically prepared copies of a quantum system, gathering statistical outcomes to a set of frequencies $\mathbf{f}$, as shown in Figure 1a. Then, the reconstruction of quantum states is realized by mapping the measured frequencies $\mathbf{f}$ into a full description of a quantum state (density matrix $\rho$) or a partial description of a quantum state (quantum properties $\boldsymbol{\mu}$) using approximation algorithms, as illustrated in Figure 1b.

Generally, to uniquely identify a quantum state, the measurements must be informatively complete to provide all the information about $\rho$ (Ježek et al., 2003), except for some special cases when density matrices of quantum states are low-rank (Haah et al., 2017) or take the form of matrix product operators (Qin et al., 2023). The exponential scaling of parameters in $\rho$ requires an exponentially increasing number of measurements, each of which requires a sufficient number of identical copies (Gebhart et al., 2023), posing a great challenge in practical applications. For example, in solid-state systems, the process associated with measuring one copy of a quantum state can be time-consuming, and implementing a sufficient number of measurement operators requires complex and

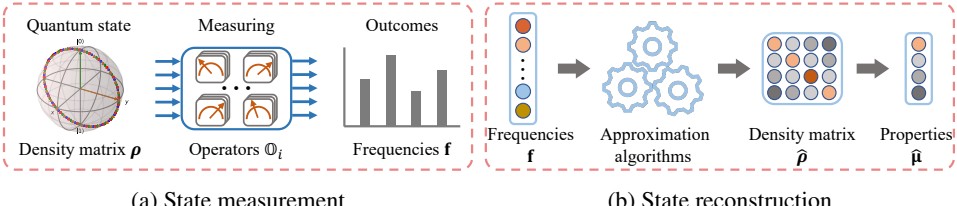

Figure 1: Schematic of QST. (a) Perform different measurements on quantum states and obtain frequencies from collected outcomes. (b) Infer quantum representations, including density matrix (full description) and quantum properties (partial description) from measured frequencies.

costly experimental setups. Given a coherence time (beyond which quantum states might change), the total copies of identical states for measurements may be constrained. In this scenario, the measurement data may be a complete but inaccurate frequency vector (a few copies assigned to each measurement operator ) or an accurate but incomplete frequency vector (Len et al., 2022) (sufficient copies assigned to partial measurement operators that are experimentally easy to generate (Chantasri et al., 2019)). The two factors are collectively referred to as imperfect scenarios (i.e., ill-posed problems) in this paper, which hinder the precise reconstruction of a quantum state.

When faced with imperfect statistics obtained in practical scenarios, traditional methods either treat those imperfections as noise in accurately measured frequencies, without employing a specific methodology (Ježek et al., 2003) or require manually-designed regularizers (Teo et al., 2012) to handle the ill-posed problem. Recently, neural network (NN) approaches have demonstrated their generality and robustness in learning intricate features from quantum systems (Gebhart et al., 2023). However, these methods have not yet incorporated imperfect factors into their network design and thus exhibit sub-optimal performance when confronted with imperfect measurement data. Practical QST suffers from noise addition and information loss in the measured frequencies, reducing the representation of measured data for accurate quantum state reconstruction. With the emergence of new architectures of NNs designed for practical problems (Vaswani et al., 2017; Devlin et al., 2018), a natural question arises: is it possible to develop an enhanced architecture that corporates imperfect measurement data to learn an informative representation from imperfect measurement data?

Recognizing the representation reduction posed by the ill-posed problem, we turn to a powerful neural architecture to extract a latent representation that might compensate for imperfect data. Specifically, we propose a transformer-based autoencoder to address the challenge of imperfect measurement data in practical QST applications. The encoder serves as an extractor to capture the informative latent representation (ILR) of quantum states from measured data. Through the state decoder, the latent representation then provides a useful feature that can be further leveraged to estimate a quantum state, including its density matrix and significant properties. However, simply training the autoencoder model with raw data might not entirely capitalize on the potential of intermediate layers to abstract meaningful latent representations from the imperfect data. Therefore, we further design a pre-training strategy that guides the encoder to learn a highly informative latent representation by augmenting a frequency decoder. The decoder retrieves the true probabilities of a complete set of measurements that provide more comprehensive information about quantum states. To ensure that the encoder's latent representation is sufficiently informative while maintaining model expressiveness predominantly in the encoder, the pre-training autoencoder is designed with an asymmetrical architecture, where the encoder is larger in scale compared to the decoder. The entire design facilitates the acquisition of an informative latent representation from imperfect measurement data, which is highly beneficial for the reconstruction of quantum states under imperfect conditions. The key contributions of our work are summarized as follows:

- We design a transformer-based autoencoder architecture, which learns to estimate an informative latent representation from imperfect measured data, benefiting the tomography of quantum states.
- We introduce a pre-training strategy to retrieve high-quality frequencies from imperfect data to relieve the ill-posedness and further enhance the expressiveness of the latent representation.
- Numerical simulations demonstrate that ILR has significant potential in reconstructing density matrices and predicting quantum properties. Experimental implementations on IBM quantum machines demonstrate the potential of ILR in compensating for noise in real quantum devices when directly using the model trained with simulated data.

## 2 RELATED WORK

**Traditional QST.** Various approaches have been proposed for estimating quantum states based on measured statistics (Qi et al., 2013). One commonly used method is the least-squares inversion that solves the inverse of linear equations which relates the measured quantities to the density matrix elements (Opatrný et al., 1997). Maximum likelihood estimation (MLE) is another popular method that searches for the state estimate by maximizing the probability of the observed data, but it involves a large number of nonlinear equations (Ježek et al., 2003). Linear regression estimation (LRE) transforms the problem into a linear model that can be efficiently solved but requires additional physical projection techniques to avoid non-physical quantum states (Qi et al., 2013; 2017). Bayesian tomography constructs a state using an integral averaging over all possible quantum states with proper weights (Huszár & Houlsby, 2012). While compressed sensing can estimate low-rank states (with a few eigenvectors) under an incomplete measurement setting (Flammia et al., 2012), determining specific measurement operators restricts its applicability in real-world scenarios.

**NN-based QST.** QST involves extracting useful information from measured statistics (Ježek et al., 2003) and has been investigated using different neural networks. For instance, fully connected networks have been applied to approximate a function that maps measured statistics into quantum parameters (Xu & Xu, 2018) and has been applied to denoise the state-preparation-and-measurement error (Palmieri et al., 2020). A convolutional neural network has been introduced to reconstruct 2-qubit quantum states from an image of measured outcomes (Lohani et al., 2020; 2021). Attention models have also been explored to capture long correlations among quantum data (Cha et al., 2021; Zhang & Di Ventra, 2022; Zhong et al., 2022). While NNs have been applied to handle incomplete measurements (Danaci et al., 2021), the method relies on a separate statistical technique to infer missing values from existing data and the decoupling of various functionalities into different models. The capability of NNs has also been demonstrated by their application in reconstructing density matrices (Ahmed et al., 2021) and predicting statistics of new measurements (Zhu et al., 2022). Compared to their methods utilizing a generative network, we design an enhanced NN architecture to learn an informative latent representation from imperfect measurement data, which benefits the reconstruction of quantum systems at different levels. Compared with classical shadow tomography which is specifically designed for predicting quantum properties (Huang et al., 2020), our approach offers a unified framework that can be utilized to reconstruct density matrices and predict properties.

**Autoencoder.** Autoencoding is a neural network technique to learn latent representations through an encoder that accepts raw input data and a decoder that reconstructs the input from the latent representation (Hinton & Zemel, 1993). It is a fundamental and powerful tool in deep learning with a wide range of applications, such as image denoising (Kingma & Welling, 2013; Ho et al., 2020) and reconstruction (Dong et al., 2015), natural language processing (Cambria & White, 2014), and others. The transformer architecture's self-attention mechanism (Vaswani et al., 2017) and pre-training technique, suitable for large dataset tasks, have popularized autoencoders in natural language processing (NLP) (Devlin et al., 2018; Raffel et al., 2020; Radford et al., 2018) and computer vision (Dosovitskiy et al., 2020; Liu et al., 2021; He et al., 2022), showing superiority to convolutional neural networks (CNNs) (LeCun et al., 1998; Brown et al., 2020). Inspired by the potential of attention layers to capture long-range correlation among their constituent qubits (Cha et al., 2021), we design a transformer-based autoencoder with a pre-training technique to reconstruct quantum states using the intermediate latent extracted from imperfect data.

## 3 METHOD

In this section, we first introduce the basic concepts of QST and then present the ill-posed challenge for QST with imperfect measurement data. Finally, a transformer-based autoencoder architecture with a pre-training strategy is proposed to learn the informative latent representation for QST.

### 3.1 PRELIMINARIES ABOUT QST

QST is a process of extracting useful information about a quantum state based on a set of measurements. For a $n$-qubit quantum system with dimension $d = 2^n$, the full mathematical representation of a quantum state can be described as a density matrix $\rho \in \mathbb{C}^{d \times d}$. Measurement operators are usually positive-operator-valued measurements (POVMs) and can be represented as a set of positive

semi-definite Hermitian matrices $\mathbb{O}_{i=1}^{M} \in \mathbb{C}^{d \times d}$ that sum to identity ($\sum_{i=1}^{M} \mathbb{O}_i = \mathbb{I}^{d \times d}$) (Nielsen & Chuang, 2010). According to Born's rule, when measuring the quantum state $\rho$ with the measurement operator $\mathbb{O}_i$, the true probability $p_i \in [0, 1]$ of obtaining the outcome $i$ is calculated as

$$p_i = \text{Tr}(\mathbb{O}_i \rho). \tag{1}$$

In practice, a finite number of identical copies of a quantum state are used for measurements. In Figure 1a, the statistical frequency for the $i$-outcome $f_i \in [0, 1]$ is collected as

$$f_i = n_i / N, \tag{2}$$

where $N \in \mathbb{R}$ denotes the total number of copies of identical states and $n_i \in \mathbb{R}$ denotes the occurrences for the outcome $i$. As the true probability $p_i$ is directly inaccessible from experimental results, a natural solution is to leverage the measured frequency vector $\mathbf{f} = [f_1, f_2, ..., f_M]^T \in \mathbb{R}^{M \times 1}$ which is a statistical approximation to the true probability vector $\mathbf{p} = [p_1, p_2, ..., p_M]^T \in \mathbb{R}^{M \times 1}$, to infer underlying information about the quantum state $\rho$. As illustrated in Figure 1b, reconstructing quantum states can be solved by approximating a function that maps the frequency vector $\mathbf{f}$ into desired representations of a quantum state, e.g., a density matrix $\rho$. In addition, it is also useful to deduce significant properties such as purity, entanglement, and entropy (Huang et al., 2020), which can be denoted as a real vector $\boldsymbol{\mu} = [\mu_1, \mu_2, ..., \mu_k]^T \in \mathbb{R}^{k \times 1}$, where $k$ denotes the number of properties.

## 3.2 THE ILL-POSED QST PROBLEM

From Eq. (2), $f_i$ is a statistical approximation to the true probability $p_i$ and the accuracy of $f_i$ depends on the number of copies of identical states available to each measurement operator. To determine a unique state, the required number of linearly independent measurement operators scales exponentially with qubit number $n$ (Ježek et al., 2003). In practical applications, the reconstruction of quantum states may be affected by imperfect conditions in two ways. i) Limited copies of identical states are assigned to each measurement operator among a complete set of measurements, and $\mathbf{f} = [f_1, f_2, ..., f_M]^T \in \mathbb{R}^{M \times 1}$ tends to be under-sampled, resulting in a large gap between $\mathbf{f}$ and $\mathbf{p}$, hindering the estimation of quantum states (Qi et al., 2017). ii) Some measurements in a complete set may not be easily realized (Liu et al., 2004) in practical quantum systems, e.g., solid-state systems. In this case, an informationally incomplete set of measurements results in the incompleteness of $\mathbf{f}$. Suppose $m$ measurements are not accessible in the experiments, and then some elements are missing or masked among $\mathbf{f}$, ending up with a new vector $\widetilde{\mathbf{f}} \in \mathbb{R}^{(M-m) \times 1}$. The incompleteness of $\widetilde{\mathbf{f}}$ fails to characterize the Hilbert space of quantum systems and therefore leads to an inaccurate reconstruction of quantum states (Danaci et al., 2021).

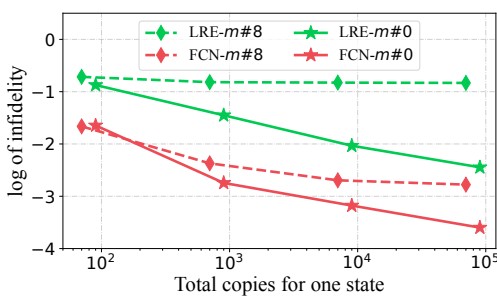

Figure 2: The ill-posed phenomena of QST. $m\#$ denotes the number of missed measurements among the cube measurements. The point with the lower value of the log of infidelity is better.

The imperfect factors discussed above correspond to an ill-posed problem, which is a common issue in machine learning. In the context of quantum estimation tasks, ill-posedness results in an inaccurate or incomplete observation of quantum states, leading to errors in subsequent QST tasks. To provide a clear explanation, we conduct an initial experiment on 2-qubit pure states using the simplest fully connected network (FCN) (Ma et al., 2021) and a traditional linear regression estimation LRE (Qi et al., 2013). The log of infidelity is measured as $\bar{F} = \log_{10}(1 - F(\rho, \hat{\rho})) \in \mathbb{R}$, where $F(\rho, \hat{\rho}) = |\text{Tr}(\sqrt{\sqrt{\rho}\hat{\rho}\sqrt{\rho}})|$ denotes the fidelity between the reconstructed state $\hat{\rho}$ and the original state $\rho$ (Nielsen & Chuang, 2010). The numerical results in Figure 2 show a strong correlation between the infidelity and the copies of the identical states or the completeness of the measurement. Specifically, we observe a sharp decrease in infidelity as the number of copies increases, while masking 8 operators among the original complete cube measurements (with a total of 36 measurements) results in a large increase in infidelity. The underlying reasons might be that the noise addition and information loss in the imperfect measurement data result in a diminished representation of quantum states. The observed ill-posed challenge promotes the necessity of acquiring an informative latent representation that can potentially compensate for the imperfections in practical QST settings.

### 3.3 LEARNING AN INFORMATIVE LATENT REPRESENTATION FOR QST WITH IMPERFECT MEASUREMENT DATA

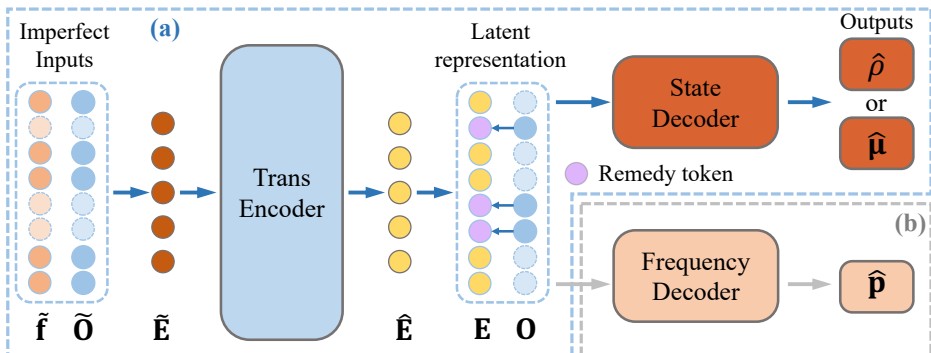

Figure 3: General schematic of learning an informative representation for quantum state tomography with imperfect measurement data. (a) The QST process aims to approximate a map from imperfect inputs (inaccurate and incomplete frequency vectors) to desired outputs (density matrices or quantum properties) through a transformer-based autoencoder and a state decoder. (b) The pre-training process aims to retrieve high-quality probabilities $\hat{\mathbf{p}}$ from masked frequency vector $\widetilde{\mathbf{f}}$ through a combination of a large encoder (share with (a)) and a small frequency decoder.

Given the representation reduction posed by ill-posed QST, it is imperative to design a robust neural architecture capable of learning a latent representation that might compensate for imperfect measurement data. Taking inspiration from the success of transformer-based autoencoders in learning latent representations from imperfect data in NLP and CV tasks, we propose an autoencoder to learn a latent representation from imperfect measurement data. The latent representation provides an informative feature that can be further leveraged to estimate quantum states by a decoder, including its density matrix and significant properties. To further enhance the informative latent representation, a pre-training process is implemented to retrieve high-quality data from imperfect measurement data by augmenting a frequency decoder. The schematic of our approach is illustrated in Figure 3.

**QST process using a transformer-based autoencoder.** In the QST process, a transformer-based autoencoder and a state decoder are combined to approximate a function that maps measurement frequencies to the desired parameters of quantum states, such as $\hat{\rho}$ (density matrix) or $\hat{\boldsymbol{\mu}}$ (physical properties). Here, we use a **hat** symbol $\hat{}$ to denote the reconstructed value from the neural networks. Hence, $\hat{\mathbf{p}}$, $\hat{\rho}$, and $\hat{\boldsymbol{\mu}}$ are utilized as the reconstructed outputs from the neural networks. In the encoding stage, we not only embed an imperfect frequency vector $\widetilde{\mathbf{f}} \in \mathbb{R}^{(M-m)\times 1}$ through a linear layer but also perform embedding on the corresponding incomplete operator matrix $\widetilde{\mathbf{O}} \in \mathbb{R}^{(M-m)\times 2d^2}$ (see **Remark** 1). Operator embedding leverages the capability of measurement operators (Ježek et al., 2003; Qi et al., 2013) in capturing the correlations between different measurements, contributing to an enhanced latent representation of quantum states (Cha et al., 2021; Zhang & Di Ventra, 2022; Zhong et al., 2022). Subsequently, these two embedded vectors are combined into a feature vector $\widetilde{\mathbf{E}} \in \mathbb{R}^{(M-m)\times L}$ (where $L$ denotes the embedding size). To reduce computational complexity by a factor of $d$, we reshape the feature vector $\widetilde{\mathbf{E}}$ into $\mathbb{R}^{G\times L}$, with $G = \frac{M-m}{d}$. Then $\widetilde{\mathbf{E}}$ is fed into a *transformer-based encoder*, which consists of multiple stacked self-attention layers and feed-forward layers. The self-attention mechanism allows the encoder to attend to different parts of the input feature vector and generate a new feature vector $\hat{\mathbf{E}} \in \mathbb{R}^{G\times L}$.

In the decoding phase, we linearly embed the missed operators $(\mathbf{O} \setminus \widetilde{\mathbf{O}}) \in \mathbb{R}^{m\times 2d^2}$, which are omitted in the encoder input, as remedy tokens added to $\hat{\mathbf{E}}$. This results in a more informative latent representation $\mathbf{E} \in \mathbb{R}^{(M/d)\times L}$, corresponding to a complete frequency vector $\mathbf{f} \in \mathbb{R}^{M\times 1}$. Finally, we inject $\mathbf{E} \in \mathbb{R}^{(M/d)\times L}$ into a *transformer-based state decoder* to deduce parameters of quantum states, including a density matrix $\hat{\rho}$ or a property vector $\hat{\boldsymbol{\mu}}$ that represents several physical features (see **Remark** 2). Taking into account the physical requirements of a density matrix, we introduce an additional post-processing module to generate a density matrix $\hat{\rho}$ from an intermediate real vector

($\hat{\nu}$) (see **Remark** 3). Finally, the parameters are trained using the mean squared error (MSE) loss between the prediction $\hat{\mu}$ and the ground-truth $\mu$, or $\hat{\nu}$ and $\nu$.

**Pre-training process using a masked and asymmetrical autoencoder.** The transformer-based encoder utilized in the QST process is specifically designed to be expressive and proficient in extracting latent features from imperfect data. However, simply training the autoencoder model with raw data may not fully utilize the potential of the intermediate layers to abstract an informative latent representation from the imperfect measurement data. A high-dimensional ILR has the capability to capture a broader range of information pertaining to quantum states, resulting in enhanced accuracy during the reconstruction process (see Figure 2). To obtain an improved ILR for QST with imperfect measurement data, we employ a pre-training strategy to guide the encoder to retrieve perfect measured data, i.e., true probabilities of a complete set of measurements. We reuse the transformer-based encoder from the QST process and combine it with a lightweight *frequency decoder* to approximate the true probabilities from imperfect frequencies.

Concretely, we draw the concept of a mask to deal with incomplete measurements and create a large number of imperfect training samples for pre-training. For example, suppose $m$ measurement operators are missing, a new frequency vector $\widetilde{\mathbf{f}} \in \mathbb{R}^{(M-m) \times 1}$ is masked out from the input frequency vector $\mathbf{f} \in \mathbb{R}^{M \times 1}$. Similarly, a corresponding mask is performed on the operator matrix $\mathbf{O} \in \mathbb{C}^{M \times 2d^2}$, ending up with a subset of the operator matrix $\widetilde{\mathbf{O}} \in \mathbb{C}^{(M-m) \times 2d^2}$. Here, the masking process can be sampled multiple times by randomly selecting $m$ from the integer set $\{0, 1, 2, 3, ..., M\}$, ending up with a large number of samples with imperfect input data. Following the transformer-based encoder reused from the QST process, we implement a specific *transformer-based frequency decoder* to approximate the original probabilities $\mathbf{p} \in \mathbb{R}^{M \times 1}$ from masked frequencies $\widetilde{\mathbf{f}} \in \mathbb{R}^{(M-m) \times 1}$, rather than using the *state decoder*. To optimize the parameters of the architecture, the pre-training process employs an MSE loss between the reconstructed probabilities $\hat{\mathbf{p}}$ and the true probabilities $\mathbf{p}$. Importantly, to prevent the expressiveness of the structure from transferring into the decoder and to ensure a sufficiently informative latent representation, the pre-training autoencoder is designed with an asymmetrical architecture, where the encoder is larger in scale compared to the decoder.

**Training strategies.** The pre-training process is first implemented to obtain a highly representative representation of quantum states. Then, complete training of the QST process is achieved by keeping the parameters of the transformer-encoder fixed and fine-tuning the parameters of the state decoder. This structure enables the preservation of the information that is learned from the pre-trained model, and the latent representation enables the state decoder to reconstruct quantum states with improved performance. To simulate different incomplete measurement scenarios, we introduce two ways for pre-training: i) *Separate*, fixed mask values are utilized for all samples, and ii) *Unified*, where mask values are randomly selected for each training batch. Similarly, the QST process has the same two training options. Please refer to **Appendix** B for their algorithmic descriptions. Now, there are four different strategies for training the whole model, including **S2S** (separate pre-training with separate tomography), **S2U** (separate pre-training with unified tomography), **U2S** (unified pre-training with separate tomography), and **U2U** (unified pre-training with unified tomography). We conduct a comprehensive comparison of 2-qubit pure states and determine **S2U** as a good strategy that balances accuracy and efficiency. Please refer to **Appendix** B for detailed information.

**Remark 1** *One operator $\mathbb{O}_i \in \mathbb{C}^{d \times d}$ can be rewritten as a real column vector $O_j \in \mathbb{R}^{2d^2 \times 1}$ by splitting its real and imaginary parts. A set of operators $\{\mathbb{O}_1, \mathbb{O}_2, ..., \mathbb{O}_M\}$ can be concatenated into an operator matrix $\mathbf{O} = [O_1, O_2, ..., O_M] \in \mathbb{R}^{M \times 2d^2}$.*

**Remark 2** *Generate a property vector $\mu$: Let the number of properties be $K$, and then multiple properties form a real vector $\mu \in \mathbb{R}^{K \times 1}$, with the ground-truth for each element calculated from the density matrix $\rho$ (Detailed information provided in **Appendix** A).*

**Remark 3** *A density matrix $\rho$ can be generated from a lower triangular matrix $\rho_L \in \mathbb{C}^{d \times d}$ according to $\rho = \rho_L \rho_L^\dagger / \mathrm{Tr}(\rho_L \rho_L^\dagger)$, satisfying (i) $\rho = \rho^\dagger$, (ii) $\mathrm{Tr}(\rho) = 1$, and (iii) $\rho \geq 0$. Conversely, any $\rho$ can be decomposed using the Cholesky decomposition (Higham, 1990), i.e., $\rho_L \rho_L^\dagger = \rho$. Furthermore, $\rho_L$ can be transformed into a real vector $\nu \in \mathbb{R}^{d^2 \times 1}$ by concatenating its real and imaginary parts. Hence, retrieving a real vector $\nu$ is equivalent to recovering the density matrix $\rho$.*

## 4 Experiments

In this section, we first provide the implementation details of ILR-QST. Then, we present the results of reconstructing density matrices as well as predicting quantum properties.

### 4.1 Implementation details

**Quantum states and measurements.** We consider both pure and mixed quantum states, where pure states can be generated by performing random unitary matrices on a pure state following the Haar metric (Danaci et al., 2021) and mixed states can be generated from the Ginibre ensembles (Forrester & Nagao, 2007) using the Hilbert-Schmidt metric (Ozawa, 2000). We adopt the cube measurement (De Burgh et al., 2008), which contains $M = 6^n$ operators for $n$-qubit systems, which is over-complete compared to the minimum number (i.e., $4^n$) of linearly independent measurements for informationally complete QST. Please refer to **Appendix** A for detailed information.

**Measured data collection.** 100,000 quantum states are randomly generated for both pure and mixed states, with 95,000 samples used for training and 5,000 for testing. Provided with states and measurement operators, we obtain true probabilities according to Eq. (1) and simulate the observation process on desktop CPUs using multinomial distribution functions with true probabilities and total copies. Upon collecting the concurrence, the frequencies are obtained according to Eq. (2).

**Neural network architecture.** The ILR-QST model is implemented on PyTorch with the following specifications: 1) Encoder: The transformer encoder consists of 8 layers with 16 attention heads (each with a dimension of 32) and 256 hidden units. The input to the encoder is a sequence of the feature latent $\tilde{E}_i$. The encoder outputs a sequence of latent features with the same dimension as the input. 2) Decoder: The parameters of the transformer model in the frequency encoder and state encoder are the same as those in the encoder, except for the number of layers. The frequency decoder employs one layer of the transformer block, whereas the state decoder consists of four transformer layers for reconstructing $\nu$ (or equivalently $\rho$) and one transformer layer for predicting $\mu$. Note that our methodology is versatile and can be employed for various numbers of qubits or different POVMs while maintaining the same architecture. The dimensions of the input and output are adapted based on the specific problem requirements. Additionally, certain hyperparameters, such as the number of layers or hidden units, may be tuned to optimize performance.

**Training settings.** The total copies are allocated into several groups, with each containing $d$ measurement operators. We define the number of copies involved in $d$ operators as $N_t$ to denote an average value. Hereafter, we use $m$ and $N_t$ to represent two imperfect factors in practical QST applications. Given a fixed $N_t$ (e.g., $N_t = 100$), the pre-train process is performed on a large number of imperfect samples, with $m$ randomly sampled from its predefined set, e.g., $\{0, 4, 8, ...\}$, while the QST process is then fine-tuned on a specific $m$ separately. In the training process, we utilize the default Adam optimizer, a batch size of 256, and an initial learning rate of $5E - 3$. To improve convergence and reduce the risk of overfitting, we employ a strategy of cosine learning rate decay. The model is trained for 500 epochs with a warm-up strategy for the first 20 epochs, during which the learning rate is gradually increased from 0 to $5E - 3$.

### 4.2 Reconstructing density matrices

Here, we implement the FCN (Ma et al., 2021), LRE (Qi et al., 2013), and MLE (Ježek et al., 2003), CNN-based method (Lohani et al., 2020) and GAN-based QST method (Ahmed et al., 2021) for comparison. To conduct an overall comparison, we compare the number of parameters, and inference time (GPU/CPU) for the proposed ILR and other methods. The simulation platform consists of an 8-core Intel(R) Xeon(R) W-2145 CPU @ 3.70GHz, 64G DDR4 memory size, and Quadro RTX 4000 with 8G. For inference time (CPU/GPU), we conduct 5,000 times of reconstructing density matrix and report the average time per inference in Table 1.

The numerical results for 2-qubit and 4-qubit pure states are summarized in Figure 4, where the log of the infidelity of the four methods increases with the number of masked operators, which is in agreement with theoretical expectations. The proposed approach (ILR) outperforms FCN and exhibits a superiority over LRE and MLE. Investigation of different number of copies ($N_t = 10$ and $N_t = 1000$) and 2-qubit mixed states are summarized in **Appendix** C. Those results suggest that the ILR is effective in reconstructing density matrices from imperfect measurement data.

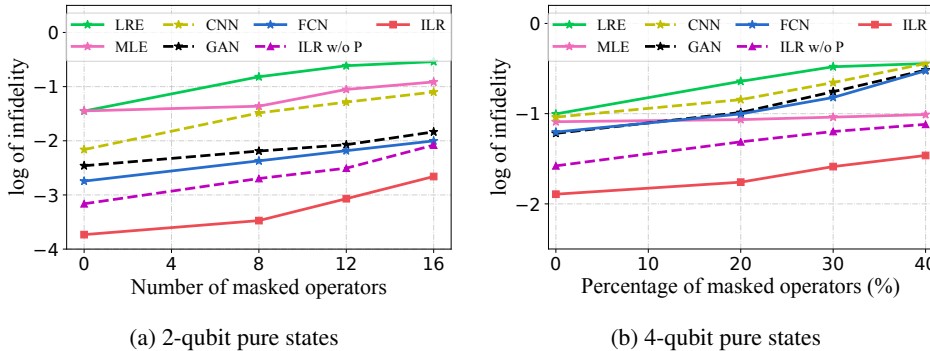

(a) 2-qubit pure states        (b) 4-qubit pure states

Figure 4: Comparison results of reconstructing density matrices for 2-qubit states ($N_t = 100$) with other methods including LRE, MLE, CNN, GAN, and FCN. "ILR w/o P" indicates ILR without the pre-training strategy.

We also perform experiments on IBM quantum machine *ibmq_manila* and results for 50 samples are provided in Table 2. Note that our ILR model is trained using simulated data that do not consider the noise model in real quantum devices. The results show the excellent robustness of our ILR model in dealing with experimental data, compared with LRE and the IBM-built method.

Table 1: Comparison of parameters and inference time for ILR and other methods used in Figure 4. Infer-CPU/GPU denotes the inference process performed on CPU/GPU. Despite the need for training and parameters in NN approaches, they consistently demonstrate superior performance to traditional methods. Notably, our ILR model outperformed FCN approaches in terms of performance with comparable parameters and inference time.

| Methods | 2 qubit | | | | 4 qubit | | | |
|---|---|---|---|---|---|---|---|---|
| | Infidelity | Params | Infer-CPU | Infer-GPU | Infidelity | Params | Infer-CPU | Infer-GPU |
| LRE | 3.54E-2 | N/A | 4.57ms | N/A | 9.93E-2 | N/A | 671ms | N/A |
| MLE | 3.56E-2 | N/A | 86.6ms | N/A | 8.14E-2 | N/A | 793ms | N/A |
| FCN | 1.80E-3 | 211K | 5.42ms | 1.30ms | 6.25E-2 | 595K | 20.35ms | 1.88ms |
| ILR | 1.85E-4 | 218K | 6.92ms | 1.38ms | 1.28E-2 | 257k | 23.71ms | 2.23ms |

Table 2: Statistical fidelities (from 50 samples) of ILR (model trained in Figure 4) and other methods on IBM quantum computer. Here a higher value of fidelity means a better performance.

| | Simulation data | | Experimental data (*ibmq_manila*) | | |
|---|---|---|---|---|---|
| | ILR | LRE | ILR | LRE | IBM-built in |
| Min | 0.999463 | 0.911296 | 0.930446 | 0.833402 | 0.819149 |
| Max | 0.999982 | 0.995680 | 0.993310 | 0.960062 | 0.957353 |
| Mean | 0.999828 | 0.972375 | 0.973660 | 0.901905 | 0.892300 |
| Variance | 2.42E-08 | 6.01E-04 | 3.16E-04 | 1.06E-03 | 1.15E-03 |

## 4.3 PREDICTING QUANTUM PROPERTIES

Apart from directly using ILR to predict quantum properties, we introduce a baseline with properties calculated from density matrices using ILR in Subsection 4.2 (termed ILR-B). The numerical results in Figure 5 together with their detailed comparsion in **Appendix** E reveal that directly predicting properties from the informative latent representation can be a good choice with good accuracy and reduced computation efforts. Then, we also summarize the results of calculating properties from reconstructed density matrices using FCN, LRE, and MLE. To distinguish them from ILR, dash lines are utilized for the above four methods. The results for 2-qubit and 4-qubit states at $N_t = 100$ are provided in Figure 5, with results of 2-qubit mixed states presented in **Appendix** D. The proposed ILR directly predicts properties with the highest accuracy, with great superiority over the indirect methods that first recover the density matrix using traditional algorithms and then calculate

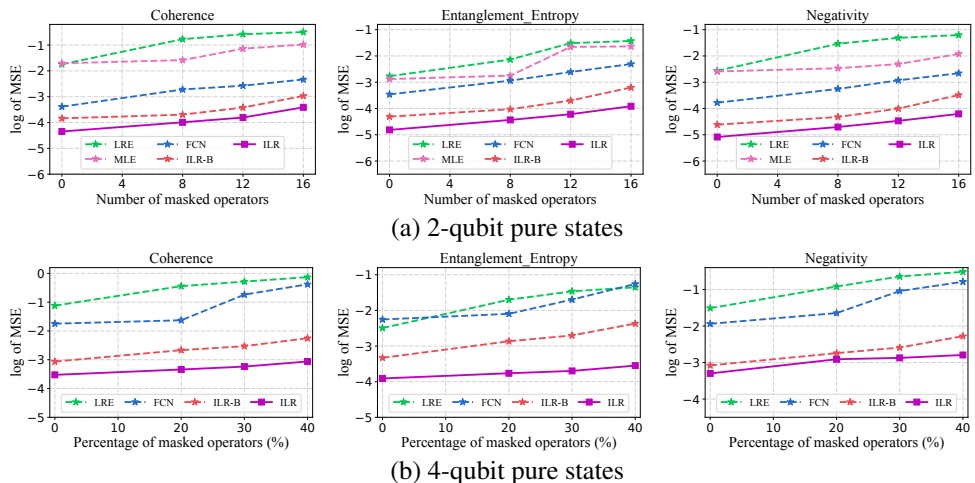

Figure 5: Results of predicting properties for pure states ($N_t = 100$). In ILR-B, FCN, MLE, and LRE (dash lines), quantum properties are calculated from the reconstructed density matrices.

properties. Those results demonstrate the informative representation has great potential to deduce a partial description of quantum states under imperfect conditions.

To demonstrate the applicability of ILR to high-dimensional states, we consider physical states with specific patterns, i.e., locally rotated GHZ states of the form $\otimes_{i=1}^{n} U_i |\psi_{GHZ}\rangle$ with $|\psi_{GHZ}\rangle = \frac{1}{\sqrt{2}}(|00...00\rangle + |11...11\rangle)$, where $U_i$ denotes a random unitary transformation in the $i$-th qubit. We also consider locally rotated W states of the form $\otimes_{i=1}^{n} U_i |\psi_W\rangle$ with $|\psi_W\rangle = \frac{1}{\sqrt{n}}(|100...00\rangle + |010...00\rangle + ... + |000...10\rangle + |000...01\rangle)$. We utilize two-qubit Pauli measurements on nearest-neighbor qubits, with $(n-1)*9*4$ types of measurement (Zhu et al., 2022). For 8 qubits, we utilize 9000 states for training and 1000 states for testing. For 12 qubits, we utilize 900 states for training and 100 states for testing. From Table 3, our method has good stability in predicting the properties of quantum states. This demonstrates that ILR has great potential to deduce a partial description of quantum states under imperfect conditions.

Table 3: MSE of predicting properties on 8/12-qubit GHZ and W states with $N_t = 100$. Mask Level (1,2,3,4): (0, 16, 24, 32) out of 63 for 8-qubit; (0, 24, 36, 48) out of 99 for 12-qubit.

| Mask Level | GHZ states (Coherence/Entanglement) | | W states (Coherence/Entanglement) | |
|---|---|---|---|---|
| | 8-qubit | 12-qubit | 8-qubit | 12-qubit |
| 1 | 2.01E-03/8.02E-04 | 1.12E-02/2.41E-03 | 8.22E-03/3.02E-03 | 7.20E-02/2.86E-03 |
| 2 | 1.26E-02/3.18E-03 | 5.36E-02/3.13E-03 | 9.37E-03/3.18E-03 | 7.19E-02/2.93E-03 |
| 3 | 1.64E-02/3.23E-03 | 6.87E-02/3.11E-03 | 1.07E-02/3.23E-03 | 7.18E-02/2.93E-03 |
| 4 | 2.44E-02/3.37E-03 | 8.82E-02/4.06E-03 | 1.25E-02/1.93E-03 | 7.23E-02/2.87E-03 |

## 5 CONCLUSION

In this paper, we investigated QST with imperfect measurement data, i.e., a few copies of identical states to approximate true probabilities or incomplete measurements that theoretically fail to specify a unique solution. Inspired by the architecture of an autoencoder that can learn latent representation from imperfect data, we designed a transformer-based autoencoder to abstract an informative latent representation from imperfect measurement data. Drawing the similarity of incompleteness and mask, we designed a masked autoencoder to map the raw measured frequencies into high-quality probabilities. By pre-training the masked autoencoder with a large number of samples obtained from ill-posed scenarios, a highly informative latent representation has been extracted and utilized to efficiently reconstruct quantum states from imperfect measured outcomes. The numerical simulations and experimental implementations demonstrate that ILR provides a robust and efficient solution for QST with imperfect measurement data.

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

## A  QUANTUM STATES, MEASUREMENTS AND PROPERTIES

**Quantum states.** In this work, we consider both pure states and mixed states. To generate pure random states, we utilize the Haar measure (Mezzadri, 2006) to generate random unitary matrices. Let $\mathbb{U}^d$ be the set of all $d$-dimensional unitary operators, i.e., any $U \in \mathbb{U}^d$ satisfies $UU^\dagger = U^\dagger U = \mathbb{I}_d$. Owing to its invariance under group multiplication (i.e., any region of $\mathcal{U}^d$ carries the same weight in a group average) (Mezzadri, 2006), the Haar metric is utilized as a probability measure on a compact group. This property enables us to generate random unitary transformations that are further utilized to generate random pure states as

$$\rho_{haar} = U|\psi_0\rangle\langle\psi_0|U^\dagger, \tag{3}$$

where $|\psi_0\rangle$ is a fixed pure state. For mixed states, we consider the random matrix from the Ginibre ensembles (Forrester & Nagao, 2007) given by $\rho_G = \mathcal{N}(0,1)_d + i\mathcal{N}(0,1)_d$, where $\mathcal{N}(0,1)_d$ represents random normal distributions of size $d \times d$ with zero mean and unity variance. Random density matrices using the Hilbert-Schmidt metric are given by (Ozawa, 2000)

$$\rho_{ginibre} = \frac{\rho_G \rho_G^\dagger}{\text{Tr}(\rho_G \rho_G^\dagger)}. \tag{4}$$

**Quantum measurements.** As for measurements, we consider tensor products of Pauli matrices, which is also called cube measurement (De Burgh et al., 2008). Denote the Pauli matrices as $\sigma = (\sigma_x, \sigma_y, \sigma_z)$, with

$$\sigma_x = \begin{bmatrix} 1 & 0 \\ 0 & -1 \end{bmatrix}, \sigma_y = \begin{bmatrix} 0 & -i \\ i & 0 \end{bmatrix}, \sigma_z = \begin{bmatrix} 0 & 1 \\ 1 & 0 \end{bmatrix}. \tag{5}$$

Let $|H\rangle$ and $|V\rangle$ be eigen-vectors of $\sigma_z$, $|L\rangle$ and $|R\rangle$ be eigen-vectors of $\sigma_y$, and $|D\rangle$ and $|A\rangle$ be eigen-vectors of $\sigma_x$. The POVM elements for one-qubit Pauli measurement are finally given as $P_{pauli} = \{|H\rangle\langle H|, |V\rangle\langle V|, |L\rangle\langle L|, |R\rangle\langle R|, |A\rangle\langle A|, |D\rangle\langle D|\}$. Generalizing the 1-qubit Pauli measurement to $n$-qubit using tensor product, we obtain the following POVMs

$$P = P_1 \otimes P_2 \cdots \otimes P_n, \quad \text{with} \quad P_i \in P_{pauli}. \tag{6}$$

For $n$-qubits, there are $6^n$ POVMs involved in the cube measurement. Typically, the $6^n$ measurement operators can be arranged into several groups, with each group containing $d = 2^n$ elements (Adamson & Steinberg, 2010). For example, $|HH\rangle\langle HH|, |HV\rangle\langle HV|, |VH\rangle\langle VH|$, and $|VV\rangle\langle VV|$ can be allocated into one group. These four POVM elements actually compose a 2-qubit detector that can be experimentally realized on quantum devices.

Table 4: Descriptions for quantum properties.

| Name | Definition | Range |
|---|---|---|
| Purity | $\text{Tr}(\rho^2)$ | $[0,1]$ |
| Entropy | $\mathbf{S}(\rho) := -\text{Tr}(\rho \ln(\rho))$ | $[0, \ln(d)]$ |
| Coherence | $S(\rho_{diag}) - S(\rho)$ | $[0, \ln(d)]$ |
| Entanglement entropy | $S(\text{Tr}_A(\rho))$ | $[0, \ln(d)/2]$ |
| Negativity | $\frac{||\rho^{\Gamma_A}||_1 - 1}{2}$ | $[0, 0.5]$ |
| Concurrence | $\max(0, \lambda_1 - \lambda_2 - \lambda_3 - \lambda_4)$ | $[0, 1]$ |

Note: $\text{Tr}_A(\rho)$ represents partial trace of $\rho$ over subsystem $A$, $\rho^{\Gamma_A}$ is partial transpose with respect to subsystem $A$. $\rho_{diag}$ denotes the matrix by deleting all the off-diagonal elements.

**Quantum properties.** In this work, we consider six properties that are widely used in quantum information processing tasks. Pure states have a fixed purity of 1 and a fixed entropy of 0, which can

be proven by simple calculation from the equations in Tabel 4. Coherence is commonly measured by the sum of squared values of non-diagonal elements among the density matrix, i.e., $\sum_{i,j,i \neq j} |\rho_{ij}|$, which can be extremely large for high-dimensional cases. Instead, we consider relative coherence regarding the entropy (Baumgratz et al., 2014). The von Neumann entanglement entropy is defined as the von Neumann entropy of either of its reduced states since they are of the same value, which can be proved from Schmidt's decomposition of the state with respect to the bipartition. Concurrence is originally defined for 2-qubit mixed states, with $\{\lambda_i\}$ being the eigenvalues (decreasing) of the Hermitian matrix $R = \sqrt{\sqrt{\rho}\rho_y\sqrt{\rho}}$ with $\rho_y = (\sigma_y \otimes \sigma_y)\rho^*(\sigma_y \otimes \sigma_y)$, with the complex conjugation $^*$ taken in the eigenbasis of the Pauli matrix $\sigma_z$. A generalized version of concurrence for multiparticle pure states in arbitrary dimension is defined as $\sqrt{2(1 - \text{Tr}(\text{Tr}_A(\rho)^2))}$. Among the six properties in Tabel 4, the latter three properties are designed to evaluate the entanglement of quantum states. Specifically, their values achieve the lowest value of zero for separable states and achieve the highest value for maximally entangled states, e.g., Greenberger–Horne–Zeilinger states.

## B  ADDITIONAL INFORMATION ABOUT TRAINING STRATEGIES

**Algorithm descriptions.** The algorithm descriptions for the pre-training process and the QST process are presented in Algorithm 1 and Algorithm 2, respectively.

---

**Algorithm 1** Description for pre-training.

**Require:** Operators $\mathbf{O}$, Frequencies $\mathbf{f}$, Ground-truth probabilities $\mathbf{p}$, Masked number list $\mathbf{m}$, Training epochs $T$, Dataset steps $S$.
**Ensure:** Obtaining the pre-training model $\Phi(\Theta_p)$.
1: Build pre-training model $\Phi(\Theta_p)$ with Encoder $\Phi(\Theta_e)$ and Frequency Decoder $\Phi(\Theta_{fd})$;
2: Initialize $\Phi(\Theta_p)$ with default Kaiming initialization;
3: **for** $t = 1, 2, \cdots, T$ **do**
4:    **for** $s = 1, 2, \cdots, S$ **do**
5:      **if** Training is *Separate* **then**
6:       Use the fixed mask number $m$;
7:      **if** Training is *Unified* **then**
8:       Random select $m$ from $\mathbf{m}$;
9:      Forward using $\widetilde{\mathbf{f}}$ and $\widetilde{\mathbf{O}}$ obtained with $m$;
10:     Backward by minimizing $MSE(\hat{\mathbf{p}}, \mathbf{p})$ in the pre-training process;
11:     Update $\Theta_e$ and $\Theta_{fd}$ with gradient descent;
12: **return** optimal $\Phi(\Theta_p)$.

---

**Algorithm 2** Description for QST.

**Require:** Operators $\mathbf{O}$, Frequencies $\mathbf{f}$, Ground-truth vectors $\boldsymbol{\nu}$ or $\boldsymbol{\mu}$, Masking list $\mathbf{m}$, Training epochs $T$, Dataset steps $S$.
**Ensure:** Obtaining the QST model $\Phi(\Theta_q)$.
1: Build QST model $\Phi(\Theta_q)$ with Encoder $\Phi(\Theta_e)$ and State Decoder $\Phi(\Theta_{sd})$;
2: Load $\Phi(\Theta_e)$ from $\Phi(\Theta_p)$, and initialize $\Phi(\Theta_{sd})$ with default Kaiming initialization;
3: **for** $t = 1, 2, \cdots, T$ **do**
4:    **for** $s = 1, 2, \cdots, S$ **do**
5:      **if** Training is *Separate* **then**
6:       Use the fixed mask number $m$;
7:      **if** Training is *Unified* **then**
8:       Random select $m$ from $\mathbf{m}$;
9:      Forward using $\widetilde{\mathbf{f}}$ and $\widetilde{\mathbf{O}}$ obtained with $m$;
10:     Backward by minimizing $MSE(\hat{\boldsymbol{\nu}}, \boldsymbol{\nu})$ or $MSE(\hat{\boldsymbol{\mu}}, \boldsymbol{\mu})$;
11:     Update $\Theta_{sd}$ with gradient descent;
12: **return** optimal $\Phi(\Theta_q)$.

---

Table 5: The $MSE(\hat{\mathbf{p}}, \mathbf{p})$ in the pre-training process on 2-qubit pure states. **S**: Separate; **U**: Unified.

| $m$ | $N_t = 10$ | | | | $N_t = 100$ | | | | $N_t = 1000$ | | | |
|---|---|---|---|---|---|---|---|---|---|---|---|---|
| | 0 | 8 | 12 | 16 | 0 | 8 | 12 | 16 | 0 | 8 | 12 | 16 |
| **S** | 3.86E-4 | 8.37E-4 | 1.94E-3 | 4.59E-3 | 3.41E-6 | 2.24E-5 | 5.49E-4 | 2.31E-3 | 5.04E-7 | 1.23E-5 | 5.31E-4 | 2.26E-3 |
| **U** | 3.67E-4 | 9.58E-4 | 2.31E-3 | 4.91E-3 | 9.28E-6 | 4.59E-5 | 7.05E-4 | 2.52E-3 | 1.15E-6 | 1.31E-5 | 6.47E-4 | 2.46E-3 |

**Training strategies in pre-training.** To evaluate the expressiveness of ILR learned during the pre-training process, we implement a comparison of two strategies ways on 2-qubit pure states, with results provided in Table 5. Generally, the *unified* strategy in the pre-training process demonstrates comparable performance to the *separate* strategy in most cases. Although the *separate* strategy shows superiority when $m = 0$, it comes at the expense of higher computational complexity. Hence, the *unified* strategy achieves a balance between computational cost and learning effectiveness.

**Training strategies in the combination of pre-training and QST.** Next, we explore the training strategies when combining pre-training and QST. Theoretical considerations suggest four possible strategies, but we exclude the **S2U** due to its impracticality and meaninglessness. We consider the

remaining three strategies with the sole transformer model (marked as Trans in the following figures) and investigate their performance on 2-qubit pure states. The numerical results are summarized in Figure 6, where **S2S** achieves the best reconstruction accuracy among the implemented three strategies, as indicated by the lowest log of infidelity. There is no significant difference between the three strategies when copies of identical states are very limited, e.g., $N_t = 10$. However, **S2S** exhibits clear superiority over other methods when $N_t = 1000$. When considering a practical situation with limited copies, for example, $N_t = 100$, **U2S** achieves a comparative performance to **S2S** with a slight disadvantage. Hence, **U2S** provides a useful strategy that balances accuracy and efficiency and we utilize it for the following experiments.

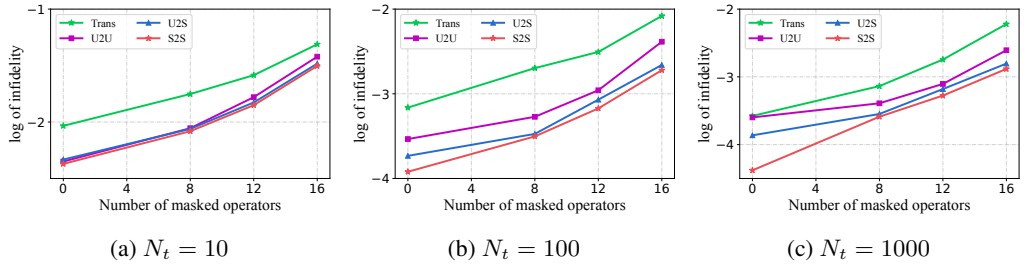

(a) $N_t = 10$      (b) $N_t = 100$      (c) $N_t = 1000$

Figure 6: Comparison of different training strategies on 2-qubit pure states.

## C ADDITIONAL RESULTS FOR RECONSTRUCTING DENSITY MATRICES

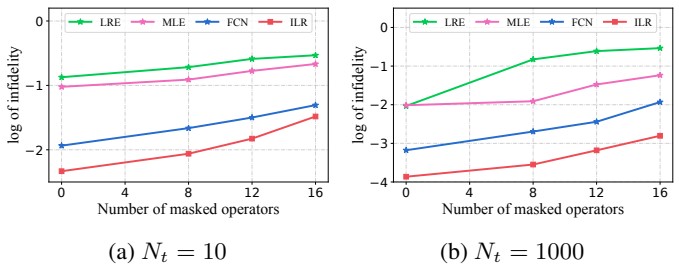

(a) $N_t = 10$      (b) $N_t = 1000$

Figure 7: Comparison results of different $N_t$ for 2-qubit pure states

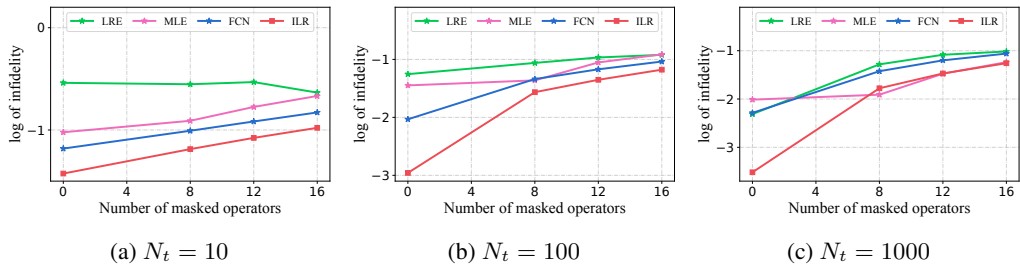

(a) $N_t = 10$      (b) $N_t = 100$      (c) $N_t = 1000$

Figure 8: Comparison results of 2-qubit mixed states.

**Numerical results with different copy numbers.** The numerical results for reconstructing density matrices of 2-qubit pure states with $N_t = 10$ and $N_t = 1000$ are summarized in Figure 7, which shows a similar trend to that of $N_t = 100$ in the main text. The proposed approach (ILR) outperforms FCN under different cases and exhibits superiority over LRE and MLE in incomplete measurements and a few copies of identical quantum states. Similar results have been observed for 2-qubit mixed states in Figure 8, where the gaps between the four methods decrease with an increasing value of $N_t$.

**ILR without operator embedding O.** To verify the function of operator embedding **O**, we have introduced an experiment that excludes operator embedding to rigorously assess the performance of ILR. The results of this experiment are presented below in Table 6. As depicted in Table 6, it is clear that the performance experiences a slight decline in the absence of operator embedding, highlighting its significant contribution.

Table 6: Infidility of reconstructing density matrices for 2-qubit states with $N_t = 100$, according to Figure 4. ILR w/o OE means ILR method without operator embedding.

| | 2-qubit pure states | | | | 4-qubit pure states | | | |
|---|---|---|---|---|---|---|---|---|
| Masked Number/Ratio | 0 | 8 | 12 | 16 | 0 | 20% | 30% | 40% |
| ILR | 1.85e-4 | 3.36e-4 | 8.52e-4 | 2.19e-3 | 1.28e-2 | 1.74e-2 | 2.59e-2 | 3.44e-2 |
| ILR w/o OE | 1.98e-4 | 5.56e-4 | 1.03e-3 | 4.19e-3 | 1.32e-2 | 2.56e-2 | 3.78e-2 | 4.78e-2 |

## D    ADDITIONAL RESULTS OF PREDICTING PROPERTIES

**2-qubit mixed states.** We focus on 6 properties of mixed states, with results summarized in Figure 9. Clearly, the proposed ILR directly predicts properties with the highest accuracy, with great superiority over other indirect methods (FCN, MLE, LRE) that recover the density matrix using traditional algorithms and then calculate properties. The proposed ILR method also surpasses the baseline (ILR-B).

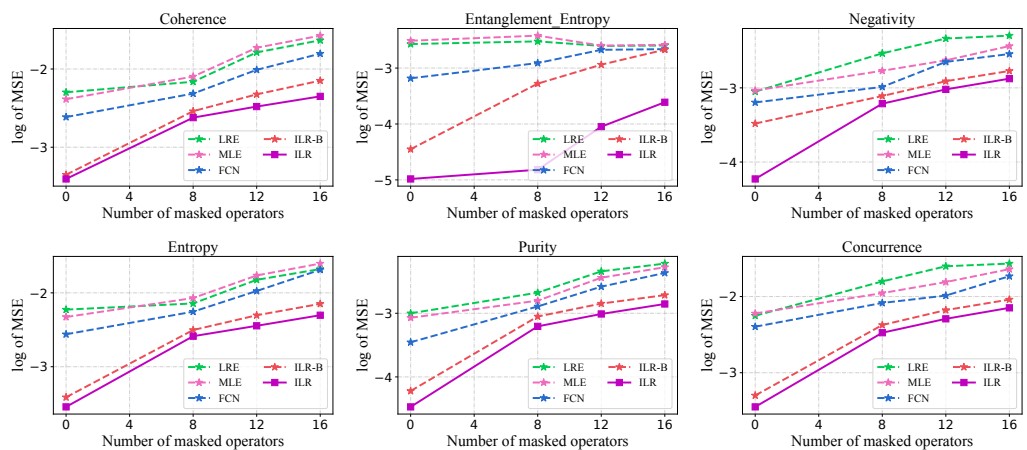

Figure 9: Comparison results of predicting properties on 2-qubit mixed states ($N_t = 100$). In ILR-B, FCN, MLE, and LRE, quantum properties are calculated from the reconstructed density matrices.

## E    DETAILED COMPARSION OF ILR AND ILR-B

Here, we compare the performance of predicting properties using ILR and the baseline that first reconstructs density matrices using ILR and then calculates properties (abbreviated as a suffix **"-B"** in the following figures). Quantum properties are abbreviated in the following figures: quantum coherence abbreviated as **"Coh"**, entanglement entropy abbreviated as **"Ent"**, negativity abbreviated as **"Neg"**, quantum entropy abbreviated as **"En"**, quantum purity abbreviated as **"Pur"** and concurrence abbreviated as **"Con"**. For example, **"Coh-B"** represents the coherence value obtained through a calculation of density matrices that are obtained from ILR.

The comparsion performance of predicting properties for pure states is summarized in Figure 10, where predicting properties without knowing their density matrices beats the baseline results under various scenarios. Similar results have been observed for predicting 6 properties of mixed states, as shown in Figure 11. A closer examination of the sub-figures reveals that the gaps of entanglement

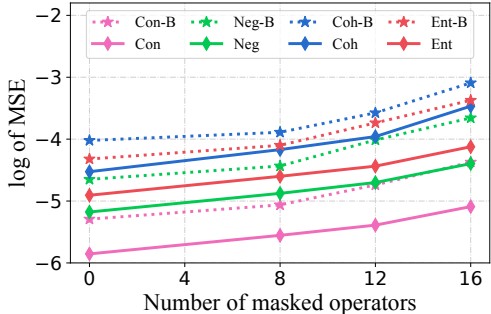

Figure 10: Two ways of property prediction for 2-qubit pure states using ILR ($N_t = 1000$). The dashed lines represent that properties are calculated from the reconstructed density matrices, while the solid lines represent that properties are directly predicted from the measured frequencies.

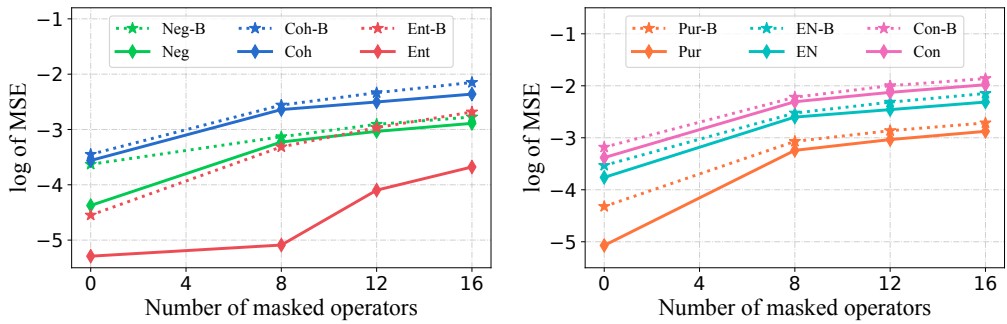

Figure 11: Two ways of property prediction for 2-qubit mixed states using ILR ($N_t = 1000$). The dashed lines represent that quantum properties are calculated from the reconstructed density matrices, while the solid lines represent that properties are predicted from the imperfect measured frequencies.

entropy between ILR and ILR-B are the most significant, indicating the effectiveness of the proposed method in predicting quantum entanglement based on imperfect measured data.

## F   RANDOM EXPERIMENTS

Table 7: Recovered infidelity on the 2-qubit pure states with our method.

| | | | | | | |
|---|---|---|---|---|---|---|
| Fixed | Seed | 1 | 10 | 20 | 30 | 40 |
| | Infidelity | 3.49E-4 | 3.84E-4 | 3.51E-4 | 3.51E-4 | 3.78E-4 |
| Random | Seed | 130 | 320 | 510 | 840 | 1011 |
| | Infidelity | 3.15E-4 | 3.61E-4 | 3.46E-4 | 3.20E-4 | 3.38E-4 |

**Random seeds.** Table 7 presents the infidelities of our method for recovering 2-qubit pure states, evaluated under different random seed settings with fewer training epochs (100) than that in the main text (500). Two scenarios are considered: fixed seed experiments and random seed experiments. For the fixed scenario, we consistently use seeds 1, 10, 20, 30, and 40, while in the random seed scenario, the seeds are randomly selected which varies substantially from 130 to 1011. The infidelities across both scenarios are in the order of $1E-4$, indicating a high fidelity in the recovered states. In the fixed seed scenario, the infidelity ranges from 3.49E-4 to 3.84E-4. In the random seed scenario, despite the larger variation in seed values, the infidelity values are comparably low, ranging from 3.15E-4 to 3.61E-4. These results suggest that our method demonstrates robust performance, maintaining

high fidelity in state recovery regardless of the specific seed value used. It is worth noting that apart from the experiments presented in Table 7, we consistently set the random seed to 1 in all other experiments in this study.

# G    CONVERGENCE RATE OF INFIDELITY FOR DIFFERENT ALGORITHMS

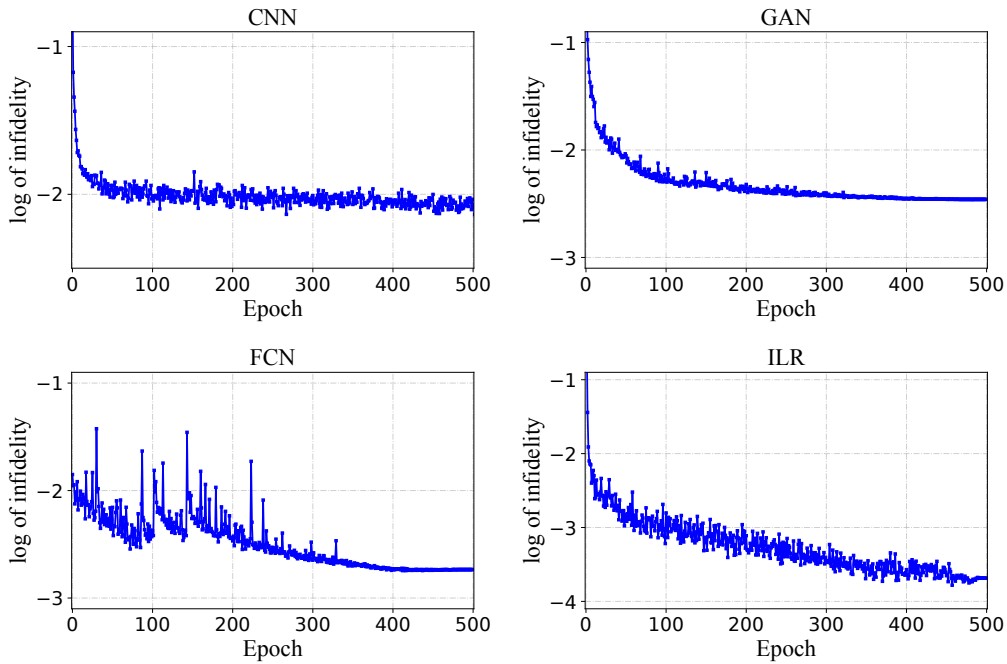

Figure 12: Convergence rate of infidelity for different algorithms, including CNN (Lohani et al., 2020), GAN (Ahmed et al., 2021), FCN (Ma et al., 2021) and ILR.

We illustrate the convergence rate of infidelity at the point with a mask number equal to 0 in Figure 4a. The convergence speed of CNN is the most rapid, achieving convergence in the early steps. GAN converges more gradually than CNN but ultimately reaches convergence. FCN displays fluctuations in the early steps but converges in later stages. ILR undergoes a continuous decrease throughout the entire training process, culminating in convergence and effectively leveraging each training iteration.

