# OpenReview forum: "Learning Informative Latent Representation for Quantum State Tomography"
_ICLR.cc/2024/Conference — ICLR 2024 Conference Withdrawn Submission_

### Official Review · Reviewer_bkkh · 2023-10-14

**Soundness:** 2 fair
**Presentation:** 3 good
**Contribution:** 2 fair
**Rating:** 5
**Confidence:** 3

**Summary:**

The paper proposes a transformer-based autoencoder architecture for quantum state tomography (QST) with imperfect measurement data. A transformer-based encoder is pre-trained to extract informative latent representation (ILR) with the task of measurement frequency reconstruction, which is succeeded by a transformer-based decoder to estimate quantum states from the measurement operators and frequencies. Extensive simulations and experiments demonstrate the remarkable ability of the proposed model to deal with imperfect measurement data in QST.

**Strengths:**

- The paper introduces a novel and interesting idea of building Transformer-based neural network for quantum state tomography.

- The paper clearly introduces the QST preliminaries, well presents the ill-posed challenge for QST and insightfully discusses the related works.

- The experiment result shows that transformer auto-encoder can reconstruct quantum states far better than the baseline models from imperfect measurement data.

**Weaknesses:**

- The idea of using transformer self-attention layers for QST is not strongly motivated, and hence not theoretically sound to me.

- The model does scales poorly with the number of qubits due to the exponential number of operators in a complete set of QST measurement, so the contribution is limited.

- The latent representation contains a mixture of encoded features and raw input features. This seems not reasonable in principle for transformer-based models, especially when the raw features and encoded features are quite different across different samples.

- The experiment is a bit slim and cannot well show the value of the proposed model.
  - It appears that the baseline models are linear regression models without pre-training, so it is an unfair comparison because the proposed model is exposed to far more data than the baseline models due to the presence of the encoder.  Are there stronger NN baselines? Is it possible to train the non-pre-training baselines with both pre-training data and training-data for this work?
  - The ablation study is missing, whereas it is necessary for this model to justify the design of different model components, such as i) having the missed operators and ii) training a frequency decoder instead of directing the state decoder in pre-training.

**Questions:**

Please kindly see the weaknesses above.

---

> ### Author Response · Authors · 2023-11-19
> **Response to Reviewer bkkh  -- Part 1**
>
> Thank you for your constructive and insightful comments for helping us improve our paper. Please see below our response to your specific questions.
>
> ---
>
> `Q1: The idea of using transformer self-attention layers for QST is not strongly motivated, and hence not theoretically sound to me.`
>
> A1: Thank you very much for your comments. Compared with other networks (such as fully connected networks and convolutional neural networks), the transformer enables to characterize the correlation between different measurements. Like sentences in natural language, entangled quantum states feature long-range correlations among their constituent qubits. Results in revised Figure 4 could verify the superiority of the transformer structure than FCN and CNN. To make it more readable, we made the following modifications:
>
> >(page 3, revised version)
>
> >`Inspired by the potential of attention layers to capture long-range correlation among their constituent qubits (Cha et. al., 2021), we design a transformer-based autoencoder` with a pre-training technique to reconstruct quantum states using the intermediate latent extracted from imperfect data.
>
>
> ---
> `Q2: The model scales poorly with the number of qubits due to the exponential number of operators in a complete set of QST measurements, so the contribution is limited.`
>
> A2: Our method aims to deal with the imperfect measurement data usually encountered in practical applications. We admit that the proposed method does not aim to solve the exponential growth in the number of measurement operators. However, our method has demonstrated its advantage over other methods under limited measurement copies or incomplete measurements.
>
> To demonstrate the effectiveness of our proposed method, we have implemented experiments on the IBM quantum devices $\{ibmq\_{manila}}$ (Qiskit-IBM-Q only supports up to 5 qubits), with results summarized in Table 2 (see Page 8, Section 4.2 in the main text). Although our ILR model is trained using simulated data that do not consider the noise model in real quantum devices, the results show excellent robustness of our ILR model in practical applications compared with LRE and the IBM-built method.
>
> In addition, we have presented the performance of predicting properties for higher qubits using two-qubit Pauli measurements on nearest-neighbor qubits, with $(n-1)*36$ measurement types, which does not introduce exponential scaling. The results in Table 1 and Table 2 (in the main text) demonstrate the scalable ability of the highly informative latent representation in predicting properties under resource-constrained conditions.
>
>
>
> ---
>
> `Q3: The latent representation contains a mixture of encoded features and raw input features.`
>
> A3: Figure 3 provides a diagrammatic sketch that may lead to confusion. In reality, the symbol $\mathbf{O}$ undergoes initial embedding through a linear layer before being added to the encoder feature. This process is elaborated on page 5, where it states, "In the decoding phase, we linearly embed the missed operators $(\mathbf{O} \setminus \mathbf{\widetilde{O}}) \in \mathbb{R}^{m\times 2d^2}$, which are omitted in the encoder input, as remedy tokens added to $\hat{\mathbf{E}}$." Therefore, it's important to note that $\mathbf{O}$ does not represent raw input features; instead, it is embedded, aligning with the standard approach in transformer-based models.
>
> ---

---

> > ### Author Response · Authors · 2023-11-19
> > **Response to Reviewer bkkh -- Part 2**
> >
> > `Q4: Experiment - It is an unfair comparison because the proposed model is exposed to far more data than the baseline models due to the presence of the encoder.`
> >
> > A4: The main contribution of this work is to design a transformer-based autoencoder architecture together with a pre-training strategy to deal with imperfect measurement data. Unlike pre-training methods employed in computer vision (CV) and natural language processing (NLP) fields, our approach involves a pre-training process with identical inputs (frequencies and operators) as the subsequent fine-tuning processes. The distinction lies in the fact that they utilize different decoders, each associated with its unique ground truth. The pre-training process aims to retrieve high-quality frequencies from imperfect data to enhance the expressiveness of the latent representation. Then, based on the informative latent representation, we introduce a state decoder to reconstruct density matrices and a property decoder to predict properties, respectively.
> >
> >
> > In the reviewed version, we implemented our ILR method without pre-training, denoted as "Trans" in Figure 6 on page 15. Now, we modify this result in Figure 4 (remarked as ILR w/o P in Figure 4, revised version) to ensure a fair comparison with other methods that do not utilize this specific strategy.
> > Moreover, to demonstrate the effectiveness of our method, we have implemented two more deep learning methods, the convolutional neural networks (CNN)-based QST [1] and GAN-based method [2] (PRL), with results provided in Figure 4 (page 8, revised version).
> > In the ill-posed measurement scenarios, the transformer-based autoencoder architecture outperforms FCN and CNN architectures. Furthermore, the incorporation of a pre-training strategy significantly enhances the performance of the autoencoder architecture.
> >
> > ---
> >
> > `Q5: Experiment - The ablation study is missing, such as i) having the missed operators and ii) training a frequency decoder instead of directing the state decoder in pre-training.`
> >
> > A5: Thank you very much for your comments. We do have some results relating to the ablation study. However, due to limited space, we present results according to different physical tasks.
> >
> > (1) The training strategies with or without pre-training are more important, so we do the related experiments in Appendix B ADDITIONAL INFORMATION ABOUT TRAIN STRATEGIES, a) we implement comparison of two strategies ways on 2-qubit pure states, with results provided in Table 5; b) We also implement ILR method without pre-training (marked as Trans) for reconstructing density matrices, as long as different pretraining and fine-tuning strategies in Figure 6.
> >
> > (2) For predicting properties, an additional comparison of these two methods is presented in APPENDIX E DETAILED COMPARISON OF ILR AND ILR-B. We have compared the way of directly predicting properties from our approach and the way of computing properties from the reconstructed density matrices using our approach (marked as ILR-B), with results provided in Figure 5.
> >
> > (3) Meanwhile, we concur with your first suggestion, which involves including the missed operators. So we have incorporated a new experiment without operator embedding to validate this adjustment, and the results (lower is better) are displayed as follows. ILR w/o OE means ILR method without operator embedding.
> >
> > | Infidility | 2 |   qubit |  pure |     states  | 4  |qubit |     pure |    states  |
> > |----------------------|----------------------|--------------------|--------------------|--------------------|----------------------|--------------------|--------------------|--------------------|
> > | Masked Number/Ratio | 0  | 8  | 12 | 16  | 0   | 20%                | 30%                | 40% |
> > | ILR   | 1.85e-4 | 3.36e-4  | 8.52e-4 | 2.19e-3  | 1.28e-2 | 1.74e-2  | 2.59e-2  | 3.44e-2  |
> > | ILR w/o OE | 1.98e-4  | 5.56e-4   | 1.03e-3  | 4.19e-3  | 1.32e-2   | 2.56e-2 | 3.78e-2 | 4.78e-2|
> >
> > (4) However, we have difficulty to understand your suggestion ii) training a frequency decoder instead of directing the state decoder in pre-training. Currently, we train an encoder and frequency decoder in the pre-training process, and fix the encoder and fine-tune a state decoder in the fine-tuning process. Could you please provide further clarification to assist us in understanding your recommendation better?
> >
> > ---
> >
> > [1] Sanjaya Lohani et al. Machine learning assisted quantum state estimation. Machine Learning: Science and Technology, 1(3): 035007, 2020.
> >
> > [2] Ahmed Shahnawaz et al. Quantum state tomography with conditional generative adversarial networks. Physical Review Letters 127.14 (2021): 140502
> >
> > ---
> >
> > We kindly request that you inform us if there are any other concerns that might have been overlooked, or if you have new ones.

---

> ### Comment · Reviewer_bkkh · 2023-11-21
>
> I would like to thank the authors for their efforts on clarifying the questions and improving the paper.  Most of my concerns have been addressed.  However, I am still not convinced of the motivation and soundness of the proposed approach, so I will raise the score to 5.
>
> Some minor points:
> - Q3: Do I understand it correctly that the missed operators are each embedded to dimension $2d^2$ and linearly projected to L-dim vector? If so, it means the original operators and missed operators are not rendered equally, and this may still lead to minor theoretical issues.
>
> -Q5: I think the same model structure can still be used to estimate the density matrix without having the missing operators in the decoder (correct me if I am wrong). I ask the question expecting to see how the model performs in comparison to pre-training (naively predicting p from f for existing operators) and prediction without involving any missing operators, because that indicates the benefit of introducing missing operators in decoder.  Similarly, the pre-trained task is different from the training task, so I think it may be helpful to compare it against pre-training with the target of estimating $\rho$ or $\mu$ (or even both) to see why it makes sense.

---

> > ### Author Response · Authors · 2023-11-22
> >
> > We appreciate your timely response and valuable comments.
> >
> > Having thoroughly reviewed your response, we think that integrating missing operators in the decoder may provide limited benefits for both reconstruction and prediction. While we haven't delved deeply into this issue, we appreciate your suggestion. Thank you, and we will persist in our efforts to validate this hypothesis.
> >
> > Regarding another aspect of your feedback, we interpret that you recommend training the ILR without employing a pre-training strategy to predict the density matrix or properties (or even both). If our understanding is correct, we find this suggestion to be highly beneficial. Partly, we have already investigated the ILR without pre-training, as illustrated in Figure 4, where we focused on density matrix reconstruction.  Furthermore, we will keep working to verify this in the scenario of predicting properties.
> >
> > Thanks again for your advice, suggestions, and feedback.
> >
> > Best regards,
> >
> > Authors

---

### Official Review · Reviewer_wK9G · 2023-10-29

**Soundness:** 3 good
**Presentation:** 3 good
**Contribution:** 2 fair
**Rating:** 5
**Confidence:** 4

**Summary:**

This manuscript introduces a transformer-based architecture designed to address the challenge of quantum state tomography with imperfect measurement data. The authors present the encoder-decoder framework of their model and illustrate a pre-training technique for the encoder, enabling it to reconstruct high-quality frequencies from imperfectly measured data. Furthermore, the authors show the model's effectiveness by employing it in the reconstruction of arbitrary 2-qubit and 4-qubit quantum states, as well as in the prediction of their properties.

**Strengths:**

- This manuscript presents a versatile model capable of simultaneously performing quantum tomography and predicting quantum properties.
- The paper introduces a pre-training strategy aimed at enhancing the robustness of the proposed model.
- This paper applies the proposed model to quantum state tomography of arbitrary quantum states rather than focusing on specific states or predefined quantum state sets.

**Weaknesses:**

- I have doubts about the scalability of the proposed model for large-scale quantum systems, especially considering the exponential growth in the number of cube operators required. This implies that the dimension of the input layer for this model would increase exponentially . If this holds true, the resulting model would become exceedingly large when applied to large-scale quantum systems.

  For another, the experiments about QST in this paper are limited to 2-qubit and 4-qubit quantum states. Even for 4-qubit pure states, when $N_t = 100$ and no operators are masked, the reconstruction fidelity is approximately $1-e^{-2} \approx 0.865$, which is not high. If this limitation is attributed to the relatively small value of $N_t$, the authors may consider conducting additional experiments on 4-qubit states (or even larger quantum system) to address this concern. I was unable to locate such experiments in the appendix, which predominantly shows a series of additional experiments conducted on 2-qubit states.

- I believe the proposed model lacks novelty in some sense. While it incorporates a transformer architecture, the fundamental encoder-decoder framework closely resembles those found in existing references [1] and [2] for quantum state tomography and quantum state learning. Furthermore, I feel that the pre-training strategy introduced here is similar to the setting in [2], which involves predicting measurement results for unmeasured bases.  I would appreciate it if the authors could clarify the distinction between "masked" operators in this paper and the unmeasured bases described in [2].

  [1] Ahmed, Shahnawaz, et al. "Quantum state tomography with conditional generative adversarial networks." *Physical Review Letters* 127.14 (2021): 140502.

  [2] Zhu, Yan, et al. "Flexible learning of quantum states with generative query neural networks." *Nature Communications* 13.1 (2022): 6222.

**Questions:**

Major concerns:

- I have stated two major concerns in the "Weakness" section above, with one relating to scalability and the other relating to novelty.

Minor questions:

- In the part of predicting quantum properties, the authors utilize locally rotated GHZ states and W states. Could the authors provide information on the range of values associated with the properties to be predicted for these two types of states?

- I have some doubts about the motivation of using the model for property prediction. Why not compute the properties directly from the predicted density matrix, especially considering that the state decoder is designed to generate the density matrix as the output?

---

> ### Author Response · Authors · 2023-11-19
> **Response to Reviewer wK9G -- Part 1**
>
> Thank you for your constructive and insightful comments for helping us improve our paper. Please see below our response to your specific questions.
>
> `Q1: The scalability of the proposed model for large-scale quantum systems.`
>
> A1: Our method aims to deal with the imperfect measurement data usually encountered in practical applications. We admit that the proposed method does not aim to solve the exponential growth in the number of measurement operators. However, our method has demonstrated its advantage over other methods under limited measurement copies or incomplete measurements. In addition, we have presented the performance of predicting properties for higher qubits using two-qubit Pauli measurements on nearest-neighbor qubits, with $(n-1)*36$ measurement types, which does not introduce exponential scaling. The results in Table 1 and Table 2 (in the main text) demonstrate the scalable ability of the highly informative latent representation in predicting properties under resource-constrained conditions.
>
> ---
>
> `Q2: The experiments about QST in this paper are limited to 2-qubit and 4-qubit quantum states. Even for 4-qubit pure states, and no operators are masked, the reconstruction fidelity is approximately $1-e^{-2} \sim 0.89$, which is not high.`
>
> A2: Thank you very much for your comments. There seems to have confusion about the definition of \textbf{log of infidelity}. In this work, all of the values regarding \textbf{log of infidelity} are actually obtained using $\log_{10}(\cdot)$ rather than $\log_e(\cdot)$. In that case, the current value for -2, actually corresponds to the fidelity value of $1-10^{-2} \sim 0.99$. By comparsion, the other methods achieve values about $1-10^{-1} \sim 0.9$. In addition, when masking 40\% of the operators, our method achieves the result better than other methods under no masking operators. Those
> results in Fig. 4 (b) demonstrate the superiority of our method in dealing with few measurement copies.
>
> To make it more readable, we have made the following modifications:
>
> >(page 4, revised version)
>
> >The log of infidelity is measured as $\bar{F}=\log_{10}(1-F(\rho,\hat{\rho}))\in \mathbb{R}$,
>
> ---
>
> `Q3: The proposed model lacks novelty in some sense. While it incorporates a transformer architecture, the fundamental encoder-decoder framework closely resembles those found in existing references [1] and [2] for quantum state tomography and quantum state learning.`
>
> A3: Thank you very much for your comments. There is a significant difference between our method and the other two methods.
>
> Compared with the method in [1], our method aims to learn a latent representation from imperfect measured data, which can be subsequently applied to different tasks, while their method is specifically designed for reconstructing density matrices. In addition, their method introduces a discriminator to distinguish between fake data and real data, while we introduce a pre-training strategy together with a shared transformer autoencoder and decoders. Compared with GAN which is difficult to train, our approach adopts a pre-training strategy to achieve a good performance in the fine-tuning stage.
>
> Compared with the method in [2], our method is a general approach that can be applied to different tasks. Their method aims to predict measured statistics of unseen measurements, which still require further steps to obtain the density matrix or properties of quantum states. Owing to the different goals, our method and the method in [2] exhibit distinctions regarding the latent representation, the architecture, and the training procedure.
>
> (1) Latent representations: Their method aims to learn a lower-dimensional representation of quantum states to predict measured statistics of new measurements. Our method aims to take advantage of the expressive ability of highly informative representation in the latent space to characterize quantum states under resource-constrained conditions.
>
> (2) Network architectures: Their method utilizes an FCN-based representation network and an LSTM-based generation network to learn the measured statistics of new measurements, while our method utilizes a transformer-encoder with global reception fields to extract intrinsic features from imperfect data and a transformer-decoder to characterize quantum states in different levels.
>
> (3) Training procedures: Their method trains the combination of the representation network and the generation network as a whole, while our method trains the model within two stages, firstly pre-training and then fine-tuning, with the shared parts trained twice. This is a common practice in the ML community to make the best use of data to bring in improved generalization for different tasks.

---

> > ### Author Response · Authors · 2023-11-19
> > **Response to Reviewer wK9G -- Part 2**
> >
> > Following your comments, we have made the following modifications:
> >
> > >(page 3, revised version)
> >
> > >...the method relies on a separate statistical technique to infer missing values from existing data and the decoupling of various functionalities into different models. `The capability of NNs has also been demonstrated by their application in reconstructing density matrices (Ahmed et al., 2021) and predicting statistics of new measurements (Zhu et al., 2022). Compared to their methods utilizing a generative network`, we design an enhanced NN architecture to learn an informative latent representation from imperfect measurement data, which benefits the reconstruction of quantum systems at different levels.
> >
> > [1] Ahmed, Shahnawaz, et al. "Quantum state tomography with conditional generative adversarial networks." Physical Review Letters 127.14 (2021): 140502.
> >
> > [2] Zhu, Yan, et al. "Flexible learning of quantum states with generative query neural networks." Nature Communications 13.1 (2022): 6222.
> >
> > ---
> >
> > `Q4: Could the authors provide information on the range of values associated with the properties to be predicted for these two types of states.`
> >
> > A4: We include the range of values associated with the properties as follows:
> > |                 | GHZ States |  -- 12-qubit | W States -- 12-qubit | -- 12-qubit |
> > |------------|------------------------|-------------------|------------------------|------------------------|
> > |                 | Coherenc               | Entanglement      | Coherence              | Entanglement           |
> > | min             | 1.0355                 | 0.5587            | 2.7643                 | 0.0037                 |
> > | max             | 3.3752                 | 0.9673            | 4.3473                 | 0.3095                 |
> > | mean            | 1.8595                 | 0.8299            | 3.3909                 | 0.1802                 |
> >
> > ---
> >
> > `Q5: Why not compute the properties directly from the predicted density matrix.`
> >
> > A5: Thank you very much for your comments. We aim to propose an informative latent representation (in the pre-training stage) that can benefit different tasks (in the fine-tuning stage). When directly predicting density matrices without knowing the density matrices, the decoder can be designed with a smaller number of units compared to the one that involves the computation of density matrices. In fact, we have implemented a way of reconstructing density matrices and computing the properties (marked as ILR-B in Figure 5). Please refer to Appendix E for more details. Their comparison reveals that directly predicting quantum properties rather than calculating from density matrices brings in better accuracy with less computation efforts.
> >
> > To make it more readable, we have made the following modifications
> >
> > >(page 8, revised version)
> >
> > >Apart from directly using ILR to predict quantum properties, we introduce a baseline with properties calculated from density matrices using ILR in Subsection 4.2 (termed ILR-B). `The numerical results in Figure 5 together with their detailed comparsion in Appendix E reveal that directly predicting properties from the informative latent representation can be a good choice with good accuracy and reduced computation efforts`.
> >
> > ---
> >
> > We kindly request that you inform us if there are any other concerns that might have been overlooked, or if you have new ones.

---

> > ### Comment · Reviewer_wK9G · 2023-11-20
> >
> > I appreciate the authors' efforts in clarifying my questions. However, I will maintain my rating.
> >
> > Scalability: As the authors mentioned in their response, their proposed model cannot solve the exponential growth in the number of measurement operators needed for QST. In other words, the current proposed model is not compatible with large-scale systems, limiting its applicability to small quantum systems.
> >
> > Novelty: While I acknowledge the difference of the proposed model and its implementation compared to previous approaches, it seems to me that the fundamental framework remains quite similar.

---

> > > ### Author Response · Authors · 2023-11-21
> > >
> > > Thank you for the timely feedback. In response to your remaining concerns, we would like to provide further clarification on them.
> > >
> > > 1. Scalability: We aim to develop a new unified framework for both full-state reconstruction and quantum property prediction to handle the ill-posed problem, arising from imperfect measurement data.  Due to exponential growth in matrix storage and measurement resources, fully characterizing arbitrary quantum states with no assumption or symmetry is generally limited to low qubits. By comparison, the estimation of high qubits means partial observations of quantum states, such as predicting the properties. We have demonstrated the power of learning an informative latent representation in both low-qubit full-state reconstruction and high-qubit property prediction within resource-constrained QST through numerical and experimental implementations.
> > >
> > > 2. Novelty: The context of the addressed issue varies, the underlying motivation diverges, the neural network architecture is entirely distinct, and training strategies differ as well. Despite these disparities, we insist that our method is different from previous mentioned approaches.
> > >
> > > We do expect that our further clarification addresses your concerns. Nevertheless, the final decision for endorsement or rejection ultimately lies with you, and we deeply respect that.
> > >
> > > Many thanks for your advice, suggestions and feedback, and wish you all the best.
> > >
> > > Best regards,
> > >
> > > Authors

---

### Official Review · Reviewer_aafB · 2023-10-30

**Soundness:** 3 good
**Presentation:** 2 fair
**Contribution:** 3 good
**Rating:** 6
**Confidence:** 4

**Summary:**

In this paper, the authors a transformer-based autoencoder architecture tailored for quantum state tomography with imperfect measurement data. However, the introduction of quantum mechanics is not explicit. In addition, some important points should be emphasized.

**Strengths:**

One significant advantage of this method is its capability to provide more comprehensive information when dealing with imperfect measurement data. By using a transformer-based encoder, it effectively extracts latent information from imperfect measurement data, improving the accuracy of quantum state estimation.

**Weaknesses:**

Please review the comments below.

**Questions:**

1. On Page 1, the authors have mentioned that, "To uniquely identify a quantum state, the measurements must be informatively complete to provide
all the information about ρ (Jeˇ zek et al., 2003). The exponential scaling of parameters in $\rho$ requires
an exponentially increasing number of measurements, each of which requires a sufficient number
of identical copies (Gebhart et al., 2023)." However, this statement is not entirely precise. When the density matrix is low-rank [1] or takes the form of a matrix product operator [2], the POVM may not be informatively complete. Consequently, when a low-dimensional structure exists within the density matrix, many traditional methods can be applied with significantly fewer repeated measurements, which is an important direction to explore compared to neural network-based approaches. The reviewer suggests that this structural aspect should be included in the introduction.

[1] J. Haah, A. Harrow, Z. Ji, X. Wu, and N. Yu, “Sample-optimal tomography of quantum states,” IEEE Transactions on Information
Theory, vol. 63, no. 9, pp. 5628–5641, 2017.

[2] Zhen Qin, Casey Jameson, Zhexuan Gong, Michael B Wakin, and Zhihui Zhu.  “Stable tomography for structured quantum states,” arXiv preprint arXiv:2306.09432, 2023.

2. In Section 3.1, PRELIMINARIES ABOUT QST, it is advisable to use the notation $2^n$ instead of just $d$. This change is necessary to establish the proper context for the definition of a qubit as introduced in Section 3.2, THE ILL-POSED QST PROBLEM. Additionally, it would be beneficial to introduce the concepts of Hermitian, positive semidefinite (PSD) structure, and unit trace in the density matrix earlier in the section for improved clarity.

3. In Figure 2, due to the missing definition of qubit, for readers without any quantum background, it is hard to compute the total number of density matrices. Consequently, the number of missed measurements will be meaningless.

4. In part "QST process using a transformer-based autoencoder", should the architecture need to be designed anew for different qubits and POVMs, the authors should underscore this requirement.

5. The reviewers suggests that the authors should add the convergence rate of infidelity for different algorithms.

6. In the section 4.2 RECONSTRUCTING DENSITY MATRICES, the use of 2-qubit and 4-qubit examples may be considered limited. It would be beneficial to include discussions involving at least 8-qubit systems for a more comprehensive analysis.

---

> ### Author Response · Authors · 2023-11-19
> **Response to Reviewer aafB -- Part 1**
>
> Thank you for your constructive and insightful comments for helping us improve our paper. Please see below our response to your specific questions.
>
> ---
>
> `Q1:  The statement about informative measurement is not entirely precise. This structural aspect should be included in the introduction with two references.`
>
> A1: Thank you very much for your constructive comments. There exist some effective methods that are tailored for special quantum states. Our method is designed for QST with limited measurements that may happen in practical QST scenarios and can be applied to general quantum states without any restrictions. Following your valuable comments, we have made the following modifications:
>
> >(page 1, revised version)
>
> >`Generally, to uniquely identify a quantum state, the measurements must be informatively complete to provide all the information about $\rho$~(Jeˇzek et al., 2003), except for the special case when density matrices of quantum states are low-rank (Haah et al., 2017) or take the form of matrix product operators (Qin et al., 2023).`
>
> ---
>
> `Q2: it is advisable to use the notation
>  $2^n$ instead of just $d$. It would be beneficial to introduce the concepts of Hermitian, positive semidefinite (PSD) structure, and unit trace in the density matrix earlier in the section for improved clarity.`
>
> A2: Thank you very much for your constructive comments. We have made the following modifications:
>
> >(page 3, revised version)
>
> >QST is a process of extracting useful information about a quantum state based on a set of measurements. `For a $n$-qubit quantum system with dimension $d=2^n$`, the full mathematical representation of a quantum state can be described as a density matrix $\rho\in \mathbb{C}^{d\times d}$.
>
> >(page 1, revised version)
>
> >In the physical world, the full description of a quantum state can be represented as a density matrix $\rho$, `i.e., a positive-definite Hermitian matrix with unit trace $\rho=\rho^{\dagger}$, $\textup{Tr}(\rho)=1$.`
>
> ---
>
> `Q3: In Figure 2, due to the missing definition of the number of qubits, for readers without any quantum background, it is hard to compute the total number of density matrices.`
>
> A3: Thank you very much for your constructive comments. We have included the definition of the number of qubits in Section 3.2, with the following modifications:
>
> >(page 3, revised version)
>
> >QST is a process of extracting useful information about a quantum state based on a set of measurements. `For a $n$-qubit quantum system with dimension $d=2^n$ `, the full mathematical representation of a quantum state can be described as a density matrix $\rho\in \mathbb{C}^{d\times d}$.
>
> In Figure 2, we perform QST on 2-qubit systems based on cube measurements with a total of 36 measurements.  To make it more readable, we have made the following modifications:
>
>
> >(page 4, revised version)
>
> >Specifically, we observe a sharp decrease in infidelity as the number of copies increases, `while masking 8 operators among the original complete cube measurements (with a total of 36 measurements) results in a large increase in infidelity.` The underlying reasons might be that the noise addition and information loss in the imperfect measurement data result in a diminished representation of quantum states.
>
> ---

---

> > ### Author Response · Authors · 2023-11-19
> > **Response to Reviewer aafB -- Part 2**
> >
> > ---
> >
> > `Q4: Should the architecture need to be designed for different qubits and POVMs, the authors should underscore this requirement.`
> >
> > A4: Our approach is a general approach that can be applied to different numbers of qubits or different POVMs. The main architecture remains the same, with the exact dimensions of the input and output adjusted according to the problem. In addition, some hyperparameters, for example, the number of layers, and the number of hidden units may also be adjusted to achieve a good performance.
> >
> > Following your constructive comments, we have made the following modifications:
> >
> > >(page 7, revised version)
> >
> > >The frequency decoder employs one layer of the transformer block, whereas the state decoder consists of four transformer layers for reconstructing $\nu$ (or equivalently $\rho$) and one transformer layer for predicting $\mu$. `Note that our methodology is versatile and can be employed for various numbers of qubits or different POVMs while maintaining the same architecture. The dimensions of the input and output are adapted based on the specific problem requirements. Additionally, certain hyperparameters, such as the number of layers or hidden units, may be tuned to optimize performance.`
> >
> > ---
> >
> > `Q5: Add the convergence rate of infidelity for different algorithms.`
> >
> > A5: Thank you very much for your comments. The convergence rates of infidelity of the involved methods are summarized in Figure 12 Appendix G (revised version). The convergence speed of CNN is the most rapid, achieving convergence in the early steps. GAN converges more gradually than CNN but ultimately reaches convergence. FCN displays fluctuations in the early steps but converges in later stages. ILR undergoes a continuous decrease throughout the entire training process, culminating in convergence and effectively leveraging each training iteration.
> >
> > ---
> >
> > `Q6: In section 4.2 RECONSTRUCTING DENSITY MATRICES, 2-qubit and 4-qubit examples may be limited. It would be beneficial to include discussions involving at least 8-qubit systems for a more comprehensive analysis.`
> >
> > A6: Thank you very much for your comments. In this work, we propose a method to deal with the imperfect measurement data that can be encountered in practical applications. The proposed method can be directly applied to full tomography of higher-qubits, but we do not implement this experiment owing to limited hardware resources. The main over-heads come from the data collection (preparing a large number of quantum states and measuring quantum states using many measurements), rather than the data analysis in our proposed method. In practical applications, reconstructing density matrices of general quantum states is generally limited to lower qubits. For example, the built-in QST function through Qiskit-IBM-Q only supports up to 5 qubits. Instead, we focus on the detailed comparison of our method and other methods under a few limited copies or incomplete measurements to demonstrate the effectiveness of our method.
> >
> > ---
> >
> > We kindly request that you inform us if there are any other concerns that might have been overlooked, or if you have new ones.

---

> > > ### Comment · Reviewer_aafB · 2023-11-22
> > >
> > > Thank you for your rebuttal. I will maintain my current score.

---

> > > > ### Author Response · Authors · 2023-11-22
> > > >
> > > > Thank you again for your constructive suggestions. Furthermore, your prompt response and positive feedback are a great encouragement for us.
> > > >
> > > > Wishing you all the best.
> > > >
> > > > Best regards,
> > > >
> > > > Authors

---

### Official Review · Reviewer_aFck · 2023-10-31

**Soundness:** 2 fair
**Presentation:** 2 fair
**Contribution:** 2 fair
**Rating:** 3
**Confidence:** 5

**Summary:**

The submission extends the concept of the masked autoencoder to enhance the sample complexity of quantum state tomography. The authors have conducted numerical simulations involving systems of up to 12 qubits to assess the performance of their proposal. Nonetheless, several statements throughout the paper and the configurations used in the numerical simulations introduce confusion, making it challenging to discern the precise contributions of the submission.

**Strengths:**

The utilization of deep learning techniques to improve quantum state tomography (QST) represents an emerging and promising field. Nevertheless, the current body of work focused on designing specialized learning models for quantum state tomography remains relatively limited. The submission effectively addresses this gap and presents intriguing results.

**Weaknesses:**

The primary weakness of the submission stems from inaccuracies in statements and the presence of confusing settings. The presence of incorrect or imprecise statements obscures the novelty and technical contributions of the proposed method. Additionally, while the authors have conducted a series of numerical simulations, the absence of a comparative analysis with state-of-the-art methods hinders our ability to gauge the practical advancements offered by the proposed method.

**Questions:**

1)  The motivation behind designing the auto-decoder structure is not entirely clear. It remains uncertain whether the authors aim to directly adapt the concept of Masked autoencoders to tackle QST tasks or if deeper insights are guiding this choice. Providing more context on this decision would enhance the submission's coherence.

2) The use of a state decoder to predict state properties appears to introduce confusion. If a user's primary interest lies in estimating specific properties, more efficient methods may be available than the proposed approach. It is essential to consider that state reconstruction, even with the inclusion of masked operations, can be resource-intensive and time-consuming.

3) The numerical simulations are limited to older methods for QST. Consequently, it remains uncertain whether the purported contributions and advantages can be effectively realized in practical applications. To establish the practicality and competitiveness of the proposed approach, a systematic examination involving a wider spectrum of advanced deep learning methods is imperative. For instance, recent studies [Ahmed, Shahnawaz, et al. "Quantum state tomography with conditional generative adversarial networks." Physical Review Letters 127.14 (2021): 140502.]  have explored the use of incomplete POVM information in conjunction with a generative adversarial learning scheme to address QST tasks, and a thorough comparative analysis with such contemporary approaches would greatly enhance the submission's value and relevance.

4) In Table 3, the authors benchmark the proposed method for estimating coherence and entanglement of GHZ and W states with 8/12 qubits. Given that this task has also been investigated in a study by Zhu et al. in 2022, a comparative study becomes imperative. The relevant results would provide valuable insights into the relative strengths and weaknesses of these two methods for the specified task.

---

> ### Author Response · Authors · 2023-11-19
> **Response to Reviewer aFck -- Part 1**
>
> Thank you for your constructive and insightful comments for helping us improve our paper. Please see below our response to your specific questions.
>
> ---
> `Q1: The motivation behind designing the auto-decoder structure is not entirely clear.`
>
> A1: Thank you very much for your comments. We drew inspiration from the implementation of mask autoencoder, but focusing on different fields and having different start points. The motivation behind this is that in practical applications, QST may be challenged by incomplete measurement settings or limited measurement copies to obtain inaccurate statistics. For example, in solid-state systems, the process associated with measuring one copy of a quantum state can be time-consuming, and implementing a sufficient number of measurement operators requires complex and costly experimental setups. Given a coherence time (beyond which quantum states may change), the total copies of identical states for measurements may be constrained. In this scenario, the measurement data may be a complete but inaccurate frequency vector (a few copies assigned to each measurement operator ) or an accurate but incomplete frequency vector (sufficient copies assigned to partial measurement operators that are experimentally easy to generate). The two factors are collectively referred to as imperfect scenarios (i.e., ill-posed problems) in this paper, which hinder the precise reconstruction of a quantum state (see Fig. 2, page 4 in the main text). Recognizing the representation reduction posed by the ill-posed problem, we turn to a powerful neural architecture to extract a latent representation that may compensate for imperfect data. Specifically, we propose a transformer-based autoencoder to address the challenge of imperfect measurement data in practical QST applications.
>
> ---
> `Q2: The use of a state decoder to predict state properties appears to introduce confusion.`
>
> A2: In this work, we aim to solve two problems using a unified framework, with the pre-training process implemented (see Fig. 3., page 5 in the main text). The pre-training process can retrieve high-quality frequencies from imperfect data to enhance the expressiveness of the latent representation. Then, based on the informative latent representation, we introduce a state decoder to reconstruct density matrices and a property decoder to predict properties, respectively. For each task, its decoder is fine-tuned separately, with the pre-trained part i.e., the encoder part shared, which can reduce the model size. We also implement experiments in Appendix E to verify that directly using a property predictor can obtain better performance than calculating the properties from the reconstructed density matrices.
>
> Additionally, we consider that in practical applications, implementing a sufficient number of measurement operators requires complex and costly experimental setups, and measuring large copies of a quantum state can be time-consuming. The masked operation serves the purpose of diminishing the total sets of measurement operators, while the pre-training strategy aims to minimize the number of measurement shots required for each state. In fact, reducing the number of measurement copies may demonstrate the effective performance of our method.
>
> ---
> `Q3: It remains uncertain whether the purported contributions and advantages can be effectively realized in practical applications.?`
>
> A3: Our method is designed for practical applications that suffer from limited measurement resources, i.e., limited copies or incomplete measurements that are usually encountered in practical applications. To demonstrate the effectiveness of our proposed method, we have implemented experiments on the IBM quantum devices $ibmq\_{manila}$, with results summarized in Table 2 (see Page 8, Section 4.2 in the main text). Although our ILR model is trained using simulated data that do not consider the noise model in real quantum devices, the results show excellent robustness of our ILR model in practical applications compared with LRE and the IBM-built method.
>
> ---

---

> ### Author Response · Authors · 2023-11-19
> **Response to Reviewer aFck -- Part 2**
>
> ---
>
> `Q4: A  systematic examination involving a wider spectrum of advanced deep learning methods is imperative. Especially GAN-based method [2]`
>
> A4: Thank you very much for your constructive comments. To demonstrate the effectiveness of our method, we have implemented two additional deep learning methods, the convolutional neural networks (CNN)-based QST [1] and GAN-based method [2] (PRL), with results provided in Figure 4 (page 8, revised version).
> Moreover, we include ILR without employing the pretraining strategy to ensure a fair comparison with other methods that do not utilize this specific strategy. In these ill-posed scenarios, the transformer-based autoencoder architecture outperforms FCN and CNN architectures. Moreover, the incorporation of a pre-training strategy significantly enhances the performance of the autoencoder architecture. Following your comments, we have made the following modifications:
>
> > (page 7, revised version)
>
> > Here, we implement the FCN (Ma et al., 2021), LRE (Qi et al., 2013), and MLE (Jeˇzek et al., 2003),
> `CNN-based method (Lohani et al., 2020) and GAN-based QST method (Ahmed et al., 2021` for comparison.
>
> ---
>
> `Q5: Compared with Zhu et al. in 2022, about GHZ and W states with 8/12 qubits.`
>
> A5: We would like to compare these two methods within the same scenario. However, despite both methods considering GHZ and W states, their applications and settings differ, making it challenging to establish uniformity. Due to the challenges associated with implementing their method in our specific scenario, accomplishing a fair reconstruction of their methods will demand additional time and effort on our part. We are still working to address this aspect.
>
>
> Before posting the comparison result, we currently analyze the differences between the two methods.
> Both our work and [Zhu et al. 2022]'s method explore the idea of extracting hidden features from imperfect measurement data to learn quantum states. However, their method aims to learn a lower-dimensional representation of quantum states from measured statistics and operators, and our method aims to learn an informative latent representation from noisy and incomplete measurements that helps solve practical problems.
>
> Specifically, the two methods exhibit the following distinctions in their implementation:
>
> 1) Latent representations: Their method aims to learn a lower-dimensional representation of quantum states to predict measured statistics of new measurements. Our method aims to take advantage of the expressive ability of highly informative representation in the latent space to characterize quantum states from imperfect measurement data.
>
> 2) Network architectures: Their method utilizes an FCN-based representation network and an LSTM-based generation network to learn the measured statistics of new measurements, while our method utilizes a transformer-encoder with global reception fields to extract intrinsic features from imperfect data and a transformer-decoder to characterize quantum states in different levels.
>
> 3) Training procedures: Their method trains the combination of the representation network and the generation network as a whole, while our method trains the model within two stages, firstly pre-training and then fine-tuning, with the shared parts trained twice. This is a common practice in the ML community to make the best use of data to bring in improved generalization for different tasks.
>
> ---
>
> [1] Sanjaya Lohani et al. Machine learning assisted quantum state estimation. Machine Learning: Science and Technology, 1(3): 035007, 2020.
>
> [2] Ahmed Shahnawaz et al. Quantum state tomography with conditional generative adversarial networks. Physical Review Letters 127.14 (2021): 140502
>
> [3] Yan Zhu et al. Flexible learning of quantum states with generative query neural networks. Nature Communications, 13
> (1):1–10, 2022.
>
> ---
>
> We kindly request that you inform us if there are any other concerns that might have been overlooked, or if you have new ones.

---

> ### Author Response · Authors · 2023-11-22
>
> Dear Reviewer aFck,
>
> I hope this message finds you well.
>
> Thanks for your dedicated time to review our work. We have taken your comments seriously and have made the necessary explanations based on your suggestions. With the imminent closure of the discussion period, time is of the essence.
>
> Given that our work received your’s less favorable scores in certain aspects, we worked diligently for a continuous span of seven days during the rebuttal period, demonstrating our unwavering commitment to addressing every facet of your feedback.
>
> We kindly request your esteemed attention to our rebuttal response. Your responsible and positive endorsement holds paramount significance during this critical phase.
>
> Wishing you all the best and awaiting your valued evaluation.
>
> Best regards,
>
> Authors

---

> ### Comment · Reviewer_aFck · 2023-11-22
>
> Thank you for your response. While I recognize your attempt to employ a single model for various quantum tasks, I remain unconvinced about employing one model for two markedly distinct types of tasks. Specifically, in quantum state tomography, the learning model needs to capture comprehensive information about the quantum state. In contrast, predicting the properties of quantum states may only require capturing partial information. Due to this concern, I maintain my score at 3.

---

> > ### Author Response · Authors · 2023-11-22
> >
> > Dear Reviewer aFck,
> >
> > I respect your decision to maintain your score. However, I disagree with your opinion that full estimation and partial estimation of quantum states are two distinct tasks with no similarities/relationships.
> >
> > (1) Both full estimation and partial estimation can be achieved using measured statistics, i.e., frequencies. Each of them can be solved by approximating a function using neural networks (which is a black box with input and output). Hence, they have intrinsic similarities both in quantum background and machine learning. In addition, generally partial description (like predicting properties) can be computed from a full description of the quantum state, i.e., density matrix.  Their relationship further motivates us to unite these two subtasks together.
> >
> > (2) In your viewpoint, these subtasks are two different tasks. However, machine learning is a powerful tool that can relate two tasks together, by learning complex patterns from data. Their similarities and relationship provide solid foundations to propose a unified framework to solve them. In machine learning, pre-training one model, and fine-tuning it to sub-tasks is popular and practical.  Despite your disagreement, our results demonstrate that the proposed approach is a useful one that can solve two sub-tasks. From my viewpoint, this is an improvement in quantum machine learning fields, at least breaking some traditional ideals.
> >
> >
> > (3) In your initial feedback, you expressed skepticism regarding the practicality of our methods in real-world scenarios, seemingly overlooking our experiments with a IBM quantum computer. Following your suggestion, we replicated GAN-based method and conducted a comparative analysis under similar conditions. The majority of your initial concerns have been appropriately addressed.
> >
> > We understand that our paper may not align with your acceptance criteria. However, we request that you provide concrete evidence substantiating your decision.
> >
> > Best regards,
> >
> > Authors.

---

> ### Comment · Reviewer_aFck · 2023-11-22
>
> Thanks for the reply. While measured statistics enable both full and partial estimation, theoretical investigations reveal an exponential gap in the required number of measurements for achieving full versus partial estimation within a tolerable error. It is implausible for deep learning models to bridge this gap unless: 1) the authors claim that deep learning can efficiently and accurately simulate the dynamics of quantum computers; or 2) the explored states are highly restrictive, resulting in a simple low-dimensional representation (as seen in the numerical simulation). This is why I suggested the authors adopt the setting in [Zhu et al. 2022] to assess the proposed method's performance in learning a class of ground states around 50 qubits. I am willing to reconsider my evaluation if the authors provide compelling evidence demonstrating that deep learning can achieve a polynomial number of training examples and measurements per training example with respect to the qubit number in that scale. Btw, I never discount the use of pretraining models. My primary concern centers on the specific focus on two downstream tasks.

---

> > ### Author Response · Authors · 2023-11-23
> >
> > We believe our approach can successfully address two tasks simultaneously, and we have already endeavored to showcase this to the best of our ability.
> >
> > Thank you for your review.
> >
> > Authors.

---

### Author Response · Authors · 2023-11-19
**General response**

Dear reviewers and meta reviewers,

We appreciate all reviewers for their valuable comments and suggestions. We've thoroughly revised our manuscript by adding more details and ablation studies as follows:

* We have enhanced the explanation of quantum state tomography in the Introduction and Related Work sections, and we have also adjusted certain concepts throughout the paper to enhance overall comprehension;

* We have implemented two more deep learning methods, the convolutional neural networks (CNN)-based QST [1] and GAN-based method [2] (PRL), with results provided in Figure 4 (page 8,
revised version);

* We have included an experiment in Appendix C to compare with ILR without operator embedding.

* We have incorporated an experiment focusing on the convergence rate of fidelity for various algorithms in Appendix G.


The changes have been highlighted using the blue font in the revised paper. We will release our code and models in the camera-ready version, and please see below our response to each reviewer. If you have any questions or suggestions, please put your comments on OpenReview.

---

[1] Sanjaya Lohani et al. Machine learning assisted quantum state estimation. Machine Learning: Science and Technology, 1(3): 035007, 2020.

[2] Ahmed Shahnawaz et al. Quantum state tomography with conditional generative adversarial networks. Physical Review Letters 127.14 (2021): 140502

---

---

### Comment · Area_Chair_sfnC · 2023-11-21
**Discussion period coming to an end, and please acknowledge the reading of rebuttals**

Dear Reviewers aFck, aafB, bkkh,

As the discussion period is coming to an end, please acknowledge that you have read the rebuttals written by the authors. Thanks.

Best wishes,
AC